# How myosin VI traps its off-state, is activated and dimerizes

Louise Canon[1], Carlos Kikuti [1], Vicente J. Planelles-Herrero [1], Tianming Lin[2], Franck Mayeux[1], Helena Sirkia[1], Young il Lee[2], Leila Heidsieck[1], Léonid Velikovsky[1], Amandine David[1], Xiaoyan Liu[2], Dihia Moussaoui[1], Emma Forest[1,3], Peter Höök[2], Karl J. Petersen[1], Tomos E. Morgan[4], Aurélie Di Cicco[5], Julia Sirés-Campos [6], Emmanuel Derivery [4], Daniel Lévy [5], Cédric Delevoye [6], H. Lee Sweeney[2] ✉ & Anne Houdusse [1] ✉

Myosin VI (Myo6) is the only minus-end directed nanomotor on actin, allowing it to uniquely contribute to numerous cellular functions. As for other nanomotors, the proper functioning of Myo6 relies on precise spatiotemporal control of motor activity via a poorly defined off-state and interactions with partners. Our structural, functional, and cellular studies reveal key features of myosin regulation and indicate that not all partners can activate Myo6. TOM1 and Dab2 cannot bind the off-state, while GIPC1 binds Myo6, releases its auto-inhibition and triggers proximal dimerization. Myo6 partners thus differentially recruit Myo6. We solved a crystal structure of the proximal dimerization domain, and show that its disruption compromises endocytosis in HeLa cells, emphasizing the importance of Myo6 dimerization. Finally, we show that the L926Q deafness mutation disrupts Myo6 auto-inhibition and indirectly impairs proximal dimerization. Our study thus demonstrates the importance of partners in the control of Myo6 auto-inhibition, localization, and activation.

Myosin motor proteins generate force and/or movement from ATP hydrolysis when associated with actin filaments. Conformational changes in the motor as it progresses from ATP hydrolysis to release of inorganic phosphate and ADP on actin are amplified into large movements via a calmodulin (CaM) or light chain binding region referred to as the "Lever arm" (Fig. 1a). To control the functions of myosin motors in cells, the ATPase activity of the motor and its ability to interact with actin must be regulated both spatially and temporally. Thirteen different classes of myosin motors serve diverse cellular functions in mammalian cells[1]. The regulation of their motor activity is however poorly characterized. A general theme for the control of motor activity is the formation of intra-molecular interactions involving the C-terminal Tail region and the Motor domain of these motors. In the active form of the motor, the Tail region interacts with itself or cellular partners. The best understood is the case of the dimeric myosin II (Myo2) class[2] and myosin V (Myo5) class[3]. In cardiac muscle, impairment in the stabilization of the myosin off-state leads to severe cardiomyopathies[4]. Whether the lack of regulation of unconventional myosins can also lead to pathology has not been demonstrated.

Perhaps the most divergent form of regulation is emerging for Class VI (Myo6), VIIa (Myo7a), and X (Myo10) myosins, which all contain regions of extended stable single alpha helices (SAH)[5]. Indeed, while they are back-folded monomers in their inactive form[6–9], these motors can self-associate to form active dimers upon activation[10–12].

[1]Structural Motility, UMR 144 CNRS/Curie Institute, PSL Research University, 26 rue d'Ulm, 75258 Paris cedex 05, France. [2]Department of Pharmacology & Therapeutics and the Myology Institute, University of Florida College of Medicine, PO Box 100267, Gainesville, Florida 32610-0267, USA. [3]École Nationale Supérieure de Chimie de Montpellier, 240 Avenue du Professeur Emile Jeanbrau, 34090 Montpellier, France. [4]MRC Laboratory of Molecular Biology, Cambridge, UK. [5]Institut Curie, Université PSL, Sorbonne Université, CNRS UMR168, Laboratoire Physico-Chimie Curie, 75005 Paris, France. [6]Structure et Compartimentation Membranaire, UMR 144 CNRS/Curie Institute, PSL Research University, 26 rue d'Ulm, 75258 Paris cedex 05, France. ✉e-mail: lsweeney@ufl.edu; anne.houdusse@curie.fr

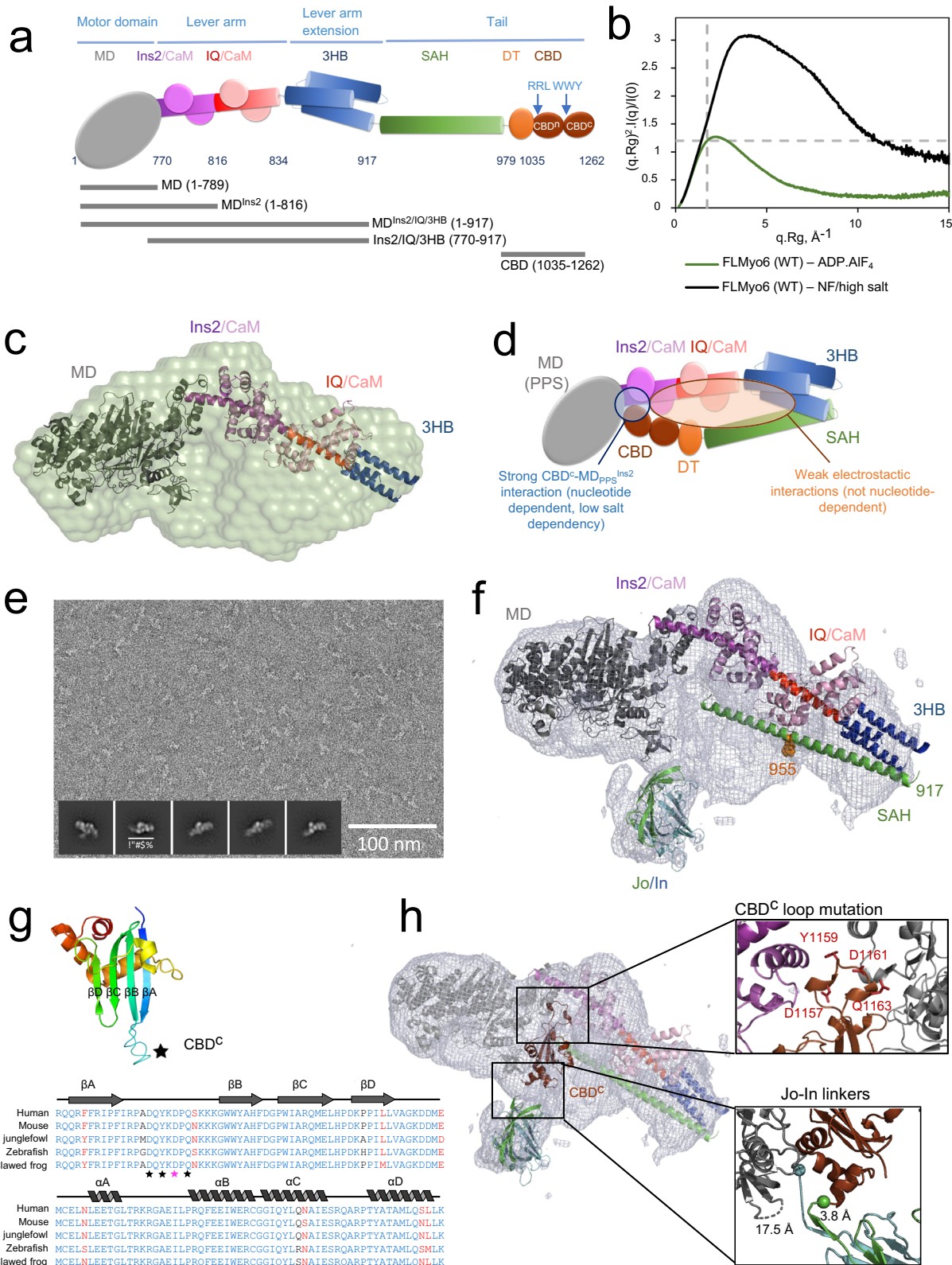

How back-folding is stabilized is unknown and the nature of the dimerization following unfolding has only been elucidated for Myo10[12], which forms an antiparallel coiled-coil immediately following the SAH. The SAH thus extends the Lever arm in the case of Myo10[12]. Whether this is also the case for the dimeric Myo6 is debated and requires elucidation of its dimerization region[11,13–18]. The manner in which the motor is dimerized and the composition of its Lever arm

greatly influence its function. A distinctive dimerization region in Myo10 allows the dimer to easily reach out for neighboring actin tracks and participate in filopodia formation[12], unlike the vesicle transporter, Myo5, which makes multiple steps on a single actin track.

As a minus-end directed actin motor, Myo6 performs unique cellular roles (reviewed in ref. 19), including endocytic vesicle trafficking and maturation, stereocilia maintenance[20] and melanosome

**Fig. 1 | Importance of ADP.P$_i$ for the compact, back-folded Myo6 conformation.**
**a** Schematic representation of FLMyo6 with the Motor domain (MD, gray), CaM binding sites (Ins2/IQ, purple/red), CaM (lilac/pink), 3-helix bundle (3HB, blue), single alpha helix (SAH, green), distal Tail (DT, orange) and CBD (brown). Residue numbers correspond to human Myo6, Uniprot entry Q9UM54-2. **b** Dimensionless Kratky plot representation from SEC-SAXS. FLMyo6 in the presence of ADP.AlF$_4$ (a widely used ADP.P$_i$ analog that stabilizes the pre-powerstroke of Myo6) (green) results in a bell-shaped spectrum with a maximum close to the intersection of the dashed lines ($\sqrt{3}$:1.104), typical of a globular protein. The spectrum for FLMyo6 in NF/high salt (black) suggests a much more elongated shape. Source data are provided as a Source Data file. **c** Representation of the ab initio SAXS envelope of Myo6 in ADP.AlF4 condition (green) with MD$^{Ins2-IQ-3HB}$ docked. Myo6 adopts a compact conformation that requires Myo6 to fold back after the 3HB domain (see "Methods" and Supplementary Fig. 3A, B). **d** Scheme representing the interactions stabilizing the Myo6 back-folded state. **e** Example of a negative staining micrograph of Jo-Myo6-In in ADP.VO$_4$ (representative of 25 grids prepared with 2 different protein batches) with selected 2D classes overlaid (from left to right: 8630; 9284; 9261; 7822 and 7179 particles averaged, respectively). **f** EM density for Jo-Myo6-In (gray mesh) obtained by negative staining. Myo6 fragments and Jo-In were manually docked inside the negative staining 3D reconstruction (see "Methods"). Negative staining 3D reconstruction and the ab initio SAXS envelope exhibit similar overall size and shape (Supplementary Fig. 3C). **g** (Top) Crystal structure of the Myo6 C-terminus (CBD$^c$) (PDB: 3H8D). Star: highly conserved and exposed loop between the β$_A$ and β$_B$ strands. (Bottom) Alignment of Myo6 CBD$^c$ domain (aa 1143 to 1262 in Q9UM54-2) from different species. Strictly conserved and similar residues are shown in blue and red, respectively. Stars: residues implicated in binding to the Myo6 Head (Table 1). **h** CBD$^c$ (brown) added to the negative staining-based model pictured in (**f**), (see "Methods"). The distances between Jo C-terminus and Myo6 N-terminus; and between Myo6 C-terminus and In N-terminus are indicated.

maturation[21], among many others. For these cellular functions, Myo6 must associate with different binding partners, such as Dab2, GIPC1 and TOM1 in distinct endosomal compartments[22–24]. Initially characterized as a deafness gene[20], Myo6 is also overexpressed in aggressive cancers[25,26] and its depletion reduces cell migration and proliferation[25,26].

Full-length Myo6 (FLMyo6) was characterized as a back-folded monomer in vitro[8,9], which was confirmed to exist in cells by FLIM (Fluorescence lifetime imaging microscopy)[27]. TOM1 and Dab2 bind Myo6 through its WWY motif on the CBD$^c$, C-terminus part of its cargo-binding domain (CBD; Fig. 1a) while GIPC1 binds the RRL motif on the CBD$^n$ (Fig. 1a). FRET studies showing that these partners can unfold constructs lacking the Myo6 Motor domain (MD) led to the proposition that all partners could activate Myo6 upon binding[27–29]. However, whether the WWY and RRL motifs are both accessible in the FLMyo6 back-folded state is unknown. Detailed studies of Myo6 recruitment are required to investigate the role of partners in the spatiotemporal regulation of its cellular activity.

The configuration of the Myo6 active state, the nature of its Lever arm and its oligomeric state can make critical differences in the way the force produced by the motor is used[16,30,31]. In fact, Myo6 is well adapted to transport as well as to anchor, depending on the load it is working against, according to single molecule assays[32]. Although the capacity of partners to favor either monomeric, dimeric or oligomeric assemblies has been described[13,27–29,31,33–35], the active configuration required to perform the different cellular roles of Myo6 is unknown. In vitro studies have identified a proximal dimerization region[11,17], but its role in the cellular function of Myo6 is not established, nor is the structure of this region. Furthermore, it is not known whether the dimerization occurs following partner binding and whether all partners lead to the same motor configuration, which ultimately will determine the nature of the effective Lever arm and mechanical performance of the motor.

A detailed description of the Myo6 off-state, a structural characterization of the proximal dimerization region, and the role of partners in Myo6 regulation are all essential to understand how Myo6 function is regulated in cells. Here we used structural and functional assays to thoroughly investigate these properties of Myo6. We demonstrate that not all partners can relieve Myo6 auto-inhibition since not all binding sites are accessible, and importantly we solved the structure of the proximal dimerization domain and demonstrate its validity.

## Results

### ADP.P$_i$ bound to the motor strongly stabilizes the off-state conformation of Myo6

Previous biophysical characterizations of the Myo6 back-folded state identified contacts between the Myo6 Lever arm and CBD (Fig. 1a)[27,36]. However, a possible role of the Motor domain in back-folding remains

to be clarified. Size Exclusion Chromatography coupled with Multi-Angle Light Scattering (SEC-MALS) and SEC coupled with Small-Angle X-ray Scattering (SEC-SAXS) experiments (Supplementary Figs. 1A–C, 2A, B; Fig. 1b) indicate that FLMyo6 adopts a compact conformation in the presence of ADP.P$_i$ analogs (Radius of gyration (Rg) = 49.23 ± 0.92 (standard deviation; SD) Å) (Fig. 1b, c; Supplementary Figs. 1C, 2A, B, 3A, B) even at high salt concentration (~425 mM NaCl) (Supplementary Fig. 1A, B). In contrast, when no nucleotide is present (nucleotide-free (NF) condition), FLMyo6 shifts from a compact to an elongated conformation in a salt concentration-dependent manner (at high salt, Rg = 84.18 ± 4.33 (SD) Å, elution 1 mL earlier from SEC-MALS) (Fig. 1b; Supplementary Fig. 1A–C, 2B). Overall, high salt dependency of FLMyo6 opening, combined with the lack of salt dependency in presence of a nucleotide, suggests that the Lever arm and the Tail are held together via electrostatic interactions, while the interactions that keep the Tail back-folded on the Head require the Motor domain to be in a nucleotide-bound state of its cycle (Fig. 1d). At very low salt (10 mM KCl) and in the presence of actin, FLMyo6 consumes ATP ~10-fold slower than the Tail-less construct MD$^{Ins2}$ (Fig. 1a; Supplementary Fig. 1D), indicating that the back-folded state is auto-inhibited.

### 3D reconstruction of the Myo6 off-state

To further characterize the Myo6 off-state, negative staining electron microscopy (EM) of FLMyo6 in ADP.VO$_4$ (ADP.Pi analog) (Supplementary Fig. 4A) resulted in heterogeneous 2D classes, likely due to the intrinsic flexibility of the protein particles. Previous FLIM demonstrated that fusion of the N- and C-termini to fluorescent proteins is compatible with Myo6 back-folding[27]. Thus, we fused the N- and C-termini of Myo6 to two covalent bonding subdomains of *Streptococcus pneumoniae* pilus adhesin RrgA: Jo and In[37] (the Jo-Myo6-In construct) to attempt to limit the inherent flexibility of the back-folded monomer.

To show that the fusion does not disrupt the Myo6 back-folding, we confirmed that the Jo-Myo6-In heavy chain could bind to two CaM using SDS-PAGE and that the Jo-Myo6-In behaved as a compact folded protein, even in NF/high salt condition using SEC-MALS and SEC-SAXS (Supplementary Figs. 4B–D, 5A). Actin-activated ATPase measurements revealed a very slow steady-state turnover rate for Jo-Myo6-In compared to earlier measurements on wild-type Myo6[38], indicating that the conformational changes required to cycle on actin were greatly slowed (Supplementary Fig. 4E). Finally, negative staining EM images of Jo-Myo6-In in ADP.VO$_4$, low salt were collected (Fig. 1e). The 3D reconstruction of the Myo6 off-state at ~17 Å resolution (Fig. 1f; Supplementary Movie 1) is consistent in shape and dimensions with SAXS data of FLMyo6 (Fig. 1c; Supplementary Fig. 3C).

### Structural model of the Myo6 off-state

The distinct EM density for the Jo-In fusion clearly defines the position of the N- and C-termini of FLMyo6 and demonstrates how it can lock

the off-state. We used available Myo6 crystallographic structures to build a model inside the 3D reconstruction (Fig. 1f, see details in "Methods"). By defining the orientation of the Lever arm, the model revealed that the flexible joint allowing back-folding must be localized around aa 912–918 prior to the SAH. The ~10 nm long SAH ends up close to the Myo6 N-terminus and Converter, where the rest of the Tail can also participate in stabilizing interactions. Importantly, only the pre-powerstroke structure of the Motor domain (which traps ADP.$P_i$, Fig. 1f), not the Rigor (NF) structure (Supplementary Fig. 4F) leads to good model-to-map agreement. We challenged this model by measuring affinities between Myo6 CBD$^{1035\text{-}end}$ and Myo6 Head fragments (Table 1, Supplementary Fig. 6). The CBD binds to the Motor domain with low affinity, and the strongest interaction ($K_D \sim 150$ nM) was measured for MD$^{Ins2/IQ/3HB}$. Removal of the IQ-3HB region (MD$^{Ins2}$) reduces the affinity by 2-fold. Last, the interaction between Myo6 CBD and MD$^{Ins2/IQ/3HB}$ drops from $K_D \sim 150$ nM to ~750 nM upon nucleotide removal (Table 1, Supplementary Fig. 6). These data indicate an interaction of the CBD with both the Motor domain and the Lever arm and highlight the importance of the nucleotide state for optimal interaction.

To define the CBD region that interacts with the MD$^{Ins2/IQ/3HB}$, we introduced four missense mutations (D1157V.Y1159D.D1161R.Q1163V: CBD$^c$ loop mutant) in a conserved and exposed loop of the CBD (Fig. 1g). These mutations abolished the ability of the CBD to bind to MD$^{Ins2/IQ/3HB}$, suggesting a key role of the CBD$^c$, and this specific loop, in the interaction (Table 1). When this information is used to dock the CBD$^c$, the Myo6 C-terminus is oriented toward the surface, close to the N-terminus consistent with our Jo-Myo6-In model (Fig. 1h, "Methods"). Finally, we performed a cross-linking mass spectrometry analysis of the purified FLMyo6 to validate our structural model (Supplementary Text, Supplementary Fig. 7A–C, Supplementary Table 1).

## Auto-inhibition of Myo6 and hearing loss

The back-folded model predicts that a sharp kink occurs at the junction between the 3HB and the SAH (Fig. 1f). The N-terminus region of the SAH (aa 922–935) is thus positioned alongside the 3HB and could participate in the stabilization of the Myo6 off-state via apolar residues found in its atypical sequence (Fig. 2a). The importance of the sequence following the 3HB for back-folding was characterized using the previously published Myo6 (SAHmimic) mutant[16], in which all apolar residues in the SAH were replaced by charged residues to match the i, i+4 alternance of a "perfect SAH sequence" (Fig. 2a). SEC-SAXS and SEC-MALS experiments indicated that FLMyo6 (SAHmimic) adopts an elongated conformation, even upon addition of an ADP.$P_i$ analog, confirming the importance of the residues 922–935 for stabilization of the Myo6 off-state (Fig. 2b, c and Supplementary Figs. 1C, 2C, D, 8A).

Interestingly, a missense mutation present in this region of the SAH (L926Q) leads to deafness in humans[39]. Positioned away from the Motor domain or from Tail regions involved in recruitment (Fig. 2a), the effect of the mutation on Myo6 function had remained elusive. SEC-MALS (Supplementary Fig. 8A) and SEC-SAXS experiments

(Fig. 2b, c) indicate that the L926Q mutation destabilizes the back-folded state. Both FLMyo6 (L926Q) and FLMyo6 (SAHmimic) mutants display higher ATPase rates ($2.86 \pm 0.12$ (SD) and $4.83 \pm 0.11$ (SD) s$^{-1}$.Head$^{-1}$, respectively) than the wild-type ($0.65 \pm 0.08$ (SD) s$^{-1}$.Head$^{-1}$) (Fig. 2d), which confirms the destabilization of the off-state.

We then investigated the impact of back-folding misregulation in the human pigmented melanoma cell line (MNT-1). Myo6 localizes to dot-like subdomains on the surface of pigmented melanosomes to promote membrane constriction and fission for the release of tubular carriers[21]. MNT-1 cells were transiently co-transfected with plasmids encoding (1) fluorescent components associated with pigmented melanosome $^{iRFP}$VAMP7[21] and $^{mCherry}$MST (melanosome-targeting tag)[40], and with (2) either FLMyo6 WT, SAHmimic, L926Q or Jo-Myo6-In, all fused to GFP. All $^{GFP}$Myo6 constructs localize as dots on melanosomes (Supplementary Fig. 8B), although at distinct levels (Supplementary Fig. 8C). The co-distribution of $^{GFP}$FLMyo6 (SAHmimic) or $^{GFP}$FLMyo6 (L926Q) with melanosomal components was greater than that of the $^{GFP}$FLMyo6 (WT) (~1.2-fold increase, $p \leq 0.001$, Supplementary Fig. 8C). However, the co-distribution of $^{GFP}$Jo-Myo6-In with melanosomes was reduced ~3-fold compared to $^{GFP}$FLMyo6 (Supplementary Fig. 8C) and the associated cytosolic and diffuse fluorescent signal was more readily observed (Supplementary Fig. 8B).

Collectively, these data indicate that Myo6 auto-inhibition drastically reduces endogenous recruitment to melanosomes while impairment of Myo6 back-folding can result in over-recruitment. These results highlight the importance of the 3HB-SAH region for Myo6 auto-inhibition since the deafness L926Q mutation is sufficient for over-recruitment of the motor. Thus, destabilization of the off-state can lead to pathology.

## Differential binding and activation of FLMyo6 by distinct cellular partners

We next aimed at distinguishing whether partners can bind to FLMyo6 in the back-folded state and if binding depends on the specific binding site. Partners interacting either with the RRL motif (GIPC1[34]) or the WWY motif (TOM1[35] and Dab2[13]) (Fig. 1a) were examined as our model suggests that in the FL off-state, the WWY motif of the CBD$^c$ is buried and unavailable for binding (Fig. 3a, b). We first looked at the ability of $^{His}$GIPC1 and $^{His}$TOM1 to bind FLMyo6 using an anti-His pull-down assay on purified proteins, in conditions promoting either Myo6 back-folding (addition of ADP.$VO_4$) or opening (NF, use of the SAHmimic mutant, or addition of $Ca^{2+}$ as previously proposed[33,36]) (Fig. 3d, Supplementary Fig. 9). Both TOM1 and GIPC1 were able to retain Myo6 in conditions that favor Myo6 opening. In contrast, upon ADP.$VO_4$ addition, the interaction of Myo6 with GIPC1 is maintained, but the interaction with TOM1 is weakened, suggesting that binding of TOM1 requires Myo6 opening.

Next, we assessed the ability of GIPC1, Dab2 and TOM1 to stimulate the ATPase activity of FLMyo6 (Fig. 3e), and found that GIPC1 increases the Myo6 ATPase rate in a concentration-dependent manner, while the addition of Dab2 or TOM1 has little impact. Note that partner

**Table 1 | Main contacts that stabilize the back-folded conformation**

| Head | Tail | Motor state | $K_D$ (nM) | $\Delta G°$ (kcal/mol) | $n$ |
|---|---|---|---|---|---|
| MD$^{Ins2/IQ/3HB}$ | $^{YFP}$CBD | ADP.$VO_4$ | $144 \pm 61$ | −9.3 | 3 |
| MD$^{Ins2}$ | $^{YFP}$CBD | ADP.$VO_4$ | $343 \pm 197$ | −8.8 | 3 |
| MD | $^{YFP}$CBD | ADP.$VO_4$ | $3920 \pm 1453$ | −7.4 | 4 |
| Ins2/IQ/3HB | $^{YFP}$CBD | - | $250 \pm 86$ | −9.0 | 4 |
| MD$^{Ins2/IQ/3HB}$ | $^{YFP}$CBD | NF | $726 \pm 480$ | −8.4 | 2 |
| MD$^{Ins2/IQ/3HB}$ | $^{YFP}$CBD$_{D1157V.Y1159D.D1161R.Q1163V}$ | ADP.$VO_4$ | n.b. | - | 3 |

Dissociation constant ($K_D$) ± $K_D$ confidence (with a 68% confidence using the NTAnalysis software) determined by microscale thermophoresis of Myo6 Head constructs against Myo6 Tail constructs (constructs schematized in Fig. 1a). Standard Gibbs free energy was obtained from the $K_D$ values, using the quantitative relationship $\Delta G = RT\ln(K_D)$. Microscale thermophoresis profiles are presented in Supplementary Fig. 6. Source data are provided as a Source Data file.

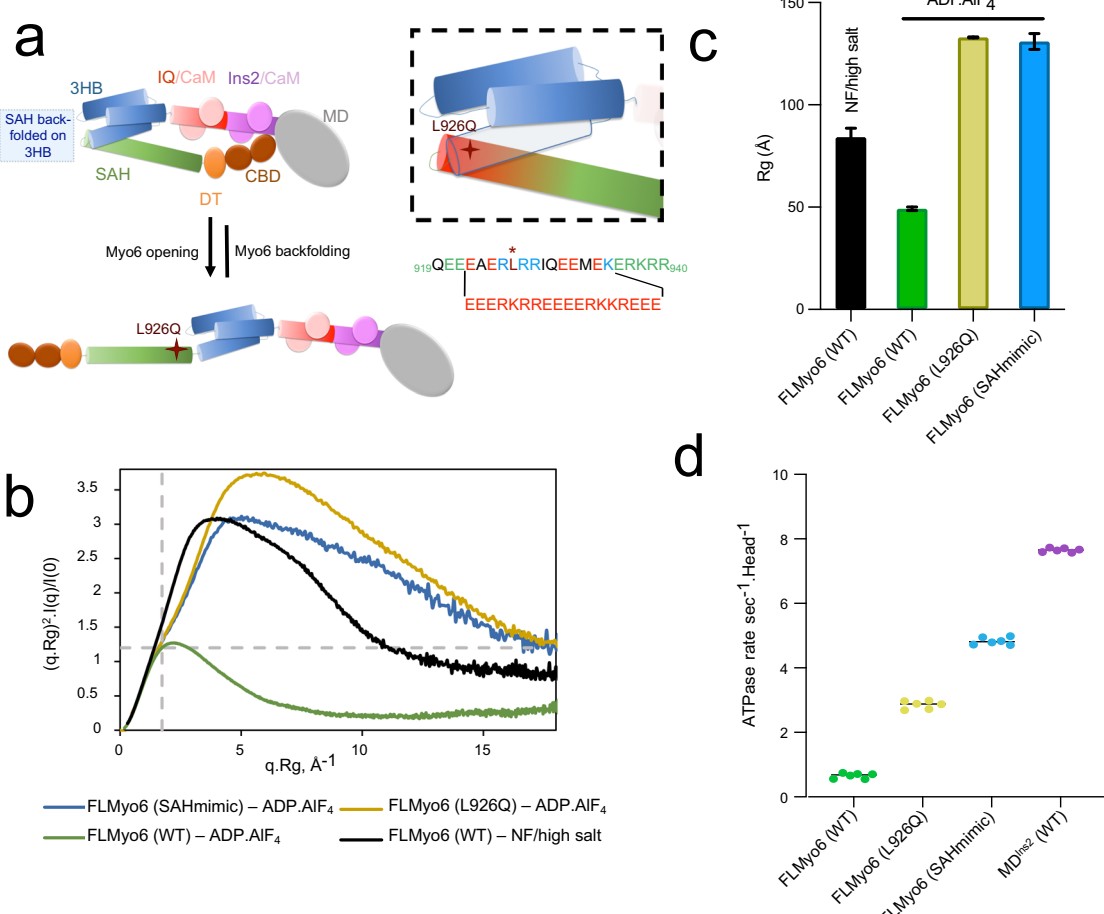

**Fig. 2 | Role of the proximal Myo6 sequence in the stabilization of the off-state.**
**a** Model of Myo6 opening/back-folding. Back-folding requires the SAH to fold back on the 3HB. The L926 residue (red cross) leads to deafness when mutated into Gln[39]. (Insert) Mutations of the apolar residues at the N-terminus of the SAH to turn Myo6 into a constitutive monomer[16] (SAHmimic). **b** Dimensionless Kratky plot representation from SEC-SAXS. FLMyo6 in NF/high salt is pictured in black. In the presence of ADP.AlF$_4$ (ADP.P$_i$ analog), FLMyo6 (L926Q) (yellow) and FLMyo6 (SAHmimic) (light blue) spectrums correspond to an elongated shape, as opposed to FLMyo6 WT (green). **c** Rg of FLMyo6 WT, L926Q and SAHmimic determined by SEC-SAXS experiments ($n = 1$) in the presence of ADP.AlF$_4$ (ADP.P$_i$ analog) and FLMyo6 in NF/high salt. Rg values were extracted from linear fits of the Guinier plots shown in Supplementary Fig. 1C using primusqt (ATSAS suite[50]). Mean ± SD. **d** Actin-activated ATPase rate of FLMyo6 WT, L926Q, SAHmimic and MD$^{Ins2}$ ($n = 6$). Mean ± SD. **(b-d)** Source data are provided as a Source Data file.

affinities for Myo6 $^{YFP}$CBD are all in the submicromolar range (i.e., sufficient to ensure binding in our ATPase assays) (Supplementary Fig. 10). Lack of activation by TOM1 and Dab2 must thus be due to inaccessibility of the WWY motif in the back-folded FLMyo6. Interestingly, TOM1 and Dab2 increase the ATPase rate of the FLMyo6 (SAHmimic) mutant, as does GIPC1, which indicates that all partners can bind to and stabilize the unfolded state (Fig. 3e).

In this context, we postulate that the RRL motif required for GIPC1 binding to CBD$^n$ (Fig. 1a) must be exposed on the surface of the back-folded Myo6, as opposed to the WWY motif. The CBD$^n$ fragment (PDB: 5V6E) was thus positioned in the unexplained density lying in continuity to CBD$^c$ in our Jo-Myo6-In 3D reconstruction (Fig. 3a–c, Supplementary Movie 1).

Collectively, these results demonstrate that not all partners can induce activation of Myo6. Partners Dab2 and TOM1 require another factor promoting Myo6 opening prior to their binding. In contrast, GIPC1 can directly activate FLMyo6, consistent with a previous study[33].

### Assessing the specific recruitment of Myo6 to native organelles by distinct partners
If some partners can relieve Myo6 from auto-inhibition, we reasoned that artificial targeting of these partners to specific cellular membranes would lead to massive recruitment of Myo6. We thus decided to artificially drive GIPC1, TOM1 and Dab2 to melanosome membranes by fusing them to the melanosome-targeting tag (MST)[40] (Supplementary Fig. 11) and verifying their ability to recruit either open (SAHmimic and L926Q) or locked (Jo-Myo6-In) Myo6. To do so with an optimized signal-to-noise measurement, we introduced the point mutation I1072A in our Myo6 constructs since it drastically reduces endogenous recruitment of Myo6 to the melanosomes ($^{GFP}$FLMyo6 (I1072A), 3.7-fold reduction ($p > 0.0001$) compared to $^{GFP}$FLMyo6; Fig. 4a; Supplementary Figs. 12A, B, 13A), while it is not part of the interface with either GIPC1 (Supplementary Fig. 10), TOM1 or Dab2. Hence, the I1072A mutation provides an easy way to reduce endogenous recruitment to melanosomes and offers a powerful tool to test the ability of distinct exogenous partners to recruit Myo6.

We transiently transfected MNT-1 cells with plasmids encoding for $^{mCherry}$MST-GIPC1, $^{mCherry}$MST-TOM1, or $^{mCherry}$MST-Dab2 for melanosome targeting. Co-transfection with plasmid encoding for $^{GFP}$Myo6 (I1072A), $^{GFP}$Jo-Myo6-In (I1072A), $^{GFP}$FLMyo6 (SAHmimic.I1072A) or $^{GFP}$FLMyo6 (L926Q.I1072A) provided a quantitative way to compare the ability of these partners to recruit Myo6 to specific organelles such as the melanosomes in cells (see "Methods").

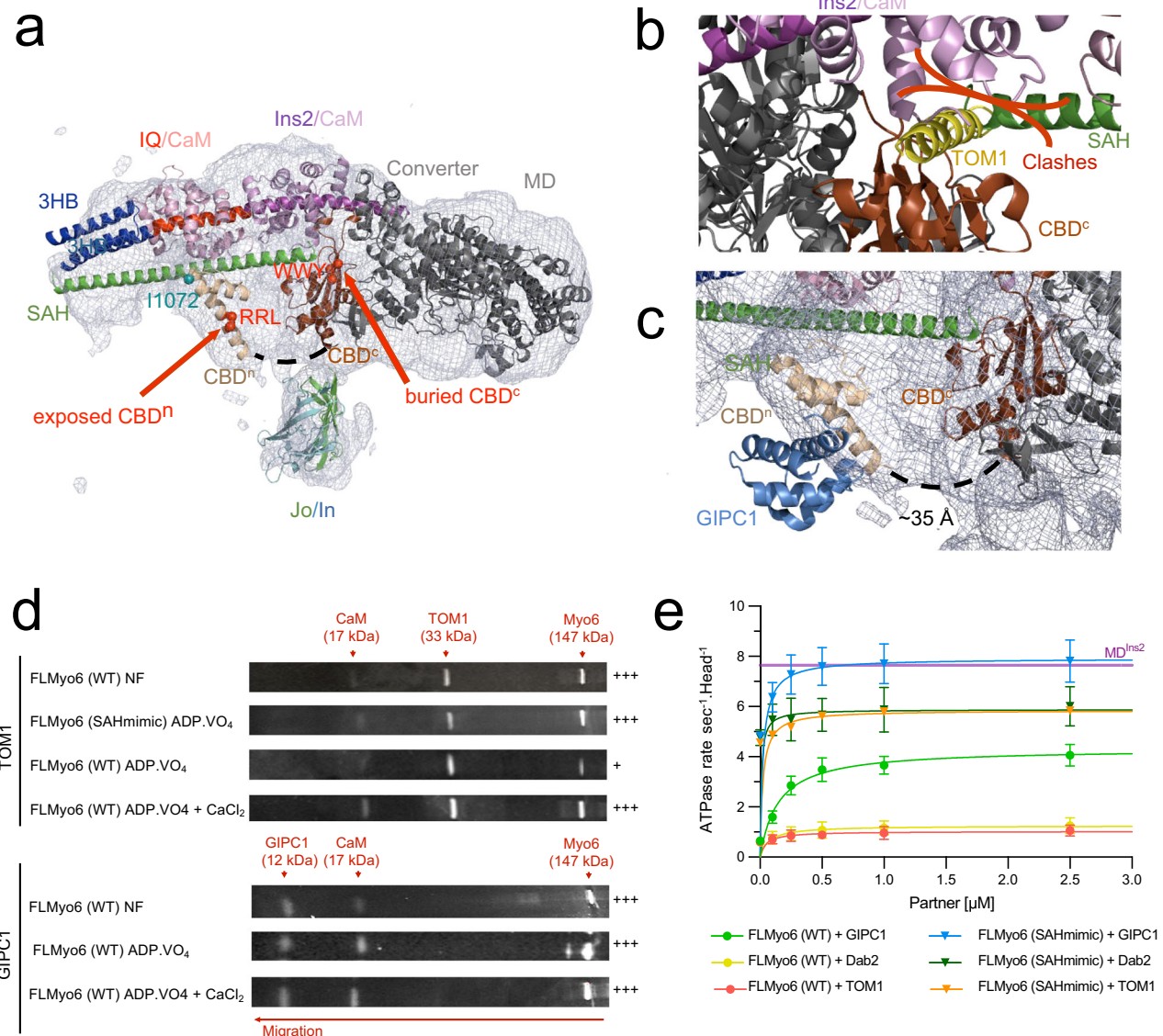

**Fig. 3 | GIPC1 can bind to and activate the back-folded form of Myo6, while Dab2 and TOM1 can only bind Myo6 once the motor has been primed open.** **a** EM density for the Jo-Myo6-In (gray mesh) obtained by negative staining, as in Fig. 1h and Supplementary Movie 1. The WWY motif (red spheres) of CBD$^c$ is buried. The CBD$^n$ fragment (beige) (PDB: 5V6E) is positioned in the remaining, uninterpreted part of the density so that the RRL motif (red spheres) on CBD$^n$ and the I1072 (blue sphere) proposed to mediate interaction between ubiquitin and Myo6[68] are both exposed. Note that no experimental model exists for 36 missing residues between the CBD$^n$ and CBD$^c$ (dashed lines) and that the position of the CBD$^n$ is consistent with the cross-links found between CBD$^n$ and the rest of the Myo6 molecule through cross-linking mass spectrometry of the purified FLMyo6 with disuccinimidyl sulfoxide (DSSO) (Supplementary Text, Supplementary Fig. 7A–C, Supplementary Table 1). The placement of elements of the Myo6 Tail within the model improved the fitting between our atomic model and the SAXS data (Supplementary Figs. 3D–F, 5B, C). **b** Fitting of CBD$^c$-TOM1 structure (PDB: 6J56) with CBD$^c$ (brown) in the model presented in Fig. 1h. TOM1 (yellow) binding would result in clashes with SAH (green) and CaM (lilac). **c** Fitting of CBD$^n$ (beige)-GIPC1 (light blue) structure (PDB: 5V6E) as for CBD$^n$ alone. GIPC1 binding seems compatible with Myo6 back-folded conformation. **d** Elutions of anti-His pull-down assays (FLMyo6 against $^{His}$TOM1 and $^{His}$GIPC1) revealed using SYPRO[69] (Input and last wash pictured in Supplementary Fig. 9). Crosses: quantification of retained Myo6 (Image-Lab software, Bio-Rad) followed by stoichiometric normalization based on partner concentration ($n = 4$ for WT and $n = 2$ for SAHmimic). + means less than 10% Myo6 retained; +++ means more than 20% Myo6 retained. **e** ATPase rates of FLMyo6 (WT) and FLMyo6 (SAHmimic) with 40 μM F-actin and increasing concentrations of GIPC1, TOM1 or Dab2 ($n = 6$). Purple line: ATPase rate of MD$^{Ins2}$ at 40 μM actin ($n = 6$) for reference. Mean ± SD. **d**, **e** Source data are provided as a Source Data file.

Expression of $^{mCherry}$MST-GIPC1 resulted in ~90% of Myo6-positive melanosomes for all the Myo6 constructs tested (Fig. 4b, e; Supplementary Fig. 13B), indicating that exogenous GIPC1 can recruit Myo6 to melanosomes independently of Myo6 being open or closed. In contrast, expression of $^{mCherry}$MST-TOM1 or $^{mCherry}$MST-Dab2 did not significantly increase the amount of $^{GFP}$Jo-Myo6-In (I1072A) positive melanosomes ($p = 0.5005$ and $p = 0.344$ respectively). This confirms the ineffectiveness of WWY partners in recruiting back-folded FLMyo6. Yet, their expression results in a 1.3/1.4-fold increase of melanosomes

containing active Myo6 mutants $^{GFP}$FLMyo6 (SAHmimic.I1072A) and $^{GFP}$FLMyo6 (L926Q.I1072A) ($p = 0.003$ or lower) (Fig. 4c–e; Supplementary Fig. 13C, D).

Interestingly, I1072A moderately affects the recruitment of Myo6 mutants impaired in auto-inhibition. Compared to $^{GFP}$FLMyo6 (SAHmimic) and $^{GFP}$FLMyo6 (L926Q), we observe reductions of 1.4 and 1.7-fold in the co-distribution with melanosome components for $^{GFP}$FLMyo6 (SAHmimic.I1072A) and $^{GFP}$FLMyo6 (L926Q.I1072A), respectively (Fig. 4a, e; Supplementary Fig. 8B, C), which are

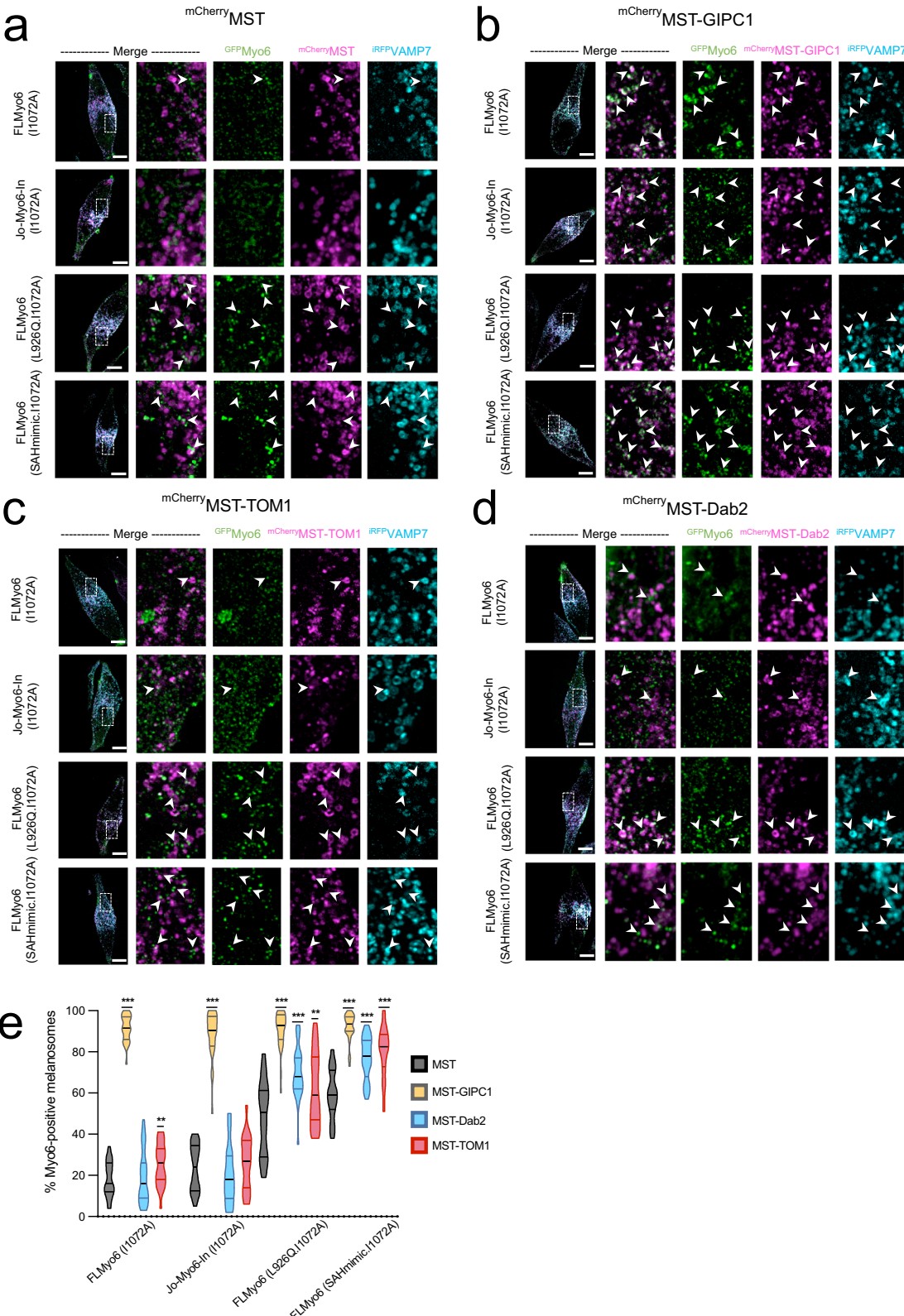

interestingly similar to the 1.6-fold reduction in recruitment observed for the CBD alone carrying the mutation I1072A (Supplementary Fig. 12C, D). We thus conclude that the I1072A mutation must reduce the affinity of the CBD for partner(s) responsible for Myo6 endogenous recruitment to melanosomes. In addition, the drastic reduction in $^{GFP}$FLMyo6 (I1072A) recruitment evidences the role of endogenous partners to promote Myo6 unfolding and indicates the major role of the I1072 residue in this process.

These results illustrate the importance of the recognition of the inactive state, and the distinct ways signaling factors can trigger association or activation of the back-folded state in a compartment for spatial and timely control of motor activity.

**Fig. 4 | GIPC1 recruits Myo6 to melanosomes independently of Myo6 closure; Dab2 and TOM1 can only recruit Myo6 after the motor has been primed open. a** Representative fixed MNT-1 cells co-expressing different <sup>GFP</sup>Myo6 (I1072A), <sup>mCherry</sup>MST and <sup>iRFP</sup>VAMP7 constructs. **b** Representative fixed MNT-1 cells co-expressing different <sup>GFP</sup>Myo6 (I1072A) constructs with <sup>mCherry</sup>MST-GIPC1 and <sup>iRFP</sup>VAMP7. **c** Representative fixed MNT-1 cells co-expressing different <sup>GFP</sup>Myo6 (I1072A) constructs with <sup>mCherry</sup>MST-TOM1 and <sup>iRFP</sup>VAMP7. **d** Representative fixed MNT-1 cells co-expressing different <sup>GFP</sup>Myo6 (I1072A) constructs with <sup>mCherry</sup>MST-Dab2 and <sup>iRFP</sup>VAMP7. (**a**–**d**) Green: Myo6 GFP; Cyan: <sup>iRFP</sup>VAMP7; Magenta: <sup>mCherry</sup>MST partner. From left to right: entire cell, 3 channels merged; 8x zoom on boxed region: <sup>GFP</sup>Myo6 / <sup>mCherry</sup>MST-partners merged, then individual channels. Scale bars: 10 μm. Arrowheads: recruitment of Myo6 on melanosomes. **e** Myo6-positive melanosomes quantification of different <sup>GFP</sup>Myo6 mutants when different <sup>mCherry</sup>MST

tagged partners are expressed (*n* = 3, total cell number-30). Myo6-positive melanosomes are expressed in percentage and normalized to the total number of VAMP7-positive melanosomes. Cells were fixed 48 h post-transfection then imaged and processed for quantification. Data are presented as the mean ± SEM. Significant stars: ***, *p* < 0.001; **, *p* < 0.01; *, *p* < 0.05; n.s., not significant (two-sided unpaired *t*-test with Welch's correction), for each <sup>GFP</sup>Myo6 construct, significance of experiments with partners compared to the control without partner (in black on the graph). *P*-values are the following: FLMyo6 (I1072A)/GIPC1: *p* < 0.0001, FLMyo6 (I1072A)/Dab2: *p* = 0.698, FLMyo6 (I1072A)/TOM1: *p* = 0.0071, Jo-Myo6-In (I1072A)/ GIPC1: *p* < 0.0001, Jo-Myo6-In (I1072A)/Dab2: *p* = 0.344, Jo-Myo6-In (I1072A) /TOM1: *p* = 0.5005, FLMyo6 (SAHmimic.I1072A)/GIPC1, TOM1 or Dab2: *p* < 0.0001, FLMyo6 (L926Q.I1072A)/GIPC1 or Dab2: *p* < 0.0001, FLMyo6 (L926Q.I1072A)/ TOM1: *p* = 0.003. Source data are provided as a Source Data file.

## A hinge that dimerizes

While we have demonstrated the key role of the sharp kink (hinge) at the 3HB/SAH junction for Myo6 auto-inhibition (Fig. 2; Supplementary Fig. 8), previous evidence by single molecule motility assays[11,16] already suggested that this region is key for Myo6 proximal dimerization. Since Myo6 proximal dimerization might be critical for a number of functional properties, we wanted to elucidate the structure of the dimerization region.

SEC-MALS with six fragments derived from the 3HB/SAH junction (Supplementary Fig. 14A, C–E) indicated that a rather conserved region (aa 875–940) can self-dimerize with $K_D^{App}$ of -19 μM ($\Delta G° \sim$ −6.4 kcal/mol) obtained from titration (Supplementary Fig. 14A, C). This minimal region corresponds to the last half of the 3HB (i.e., the 2nd and 3rd helix) and the first part of the SAH (Supplementary Fig. 14A–C). In contrast, no dimerization was observed when peptides included the whole 3HB domain, even when a peak concentration of 30 μM was reached for the 834–955 peptide (Supplementary Fig. 14A, D). This data is consistent with previous findings indicating that proximal dimerization requires unfolding of the 3HB (ref. 16, Supplementary Fig. 15A).

Crystals of the 875–940 peptide diffracted to 2.1 Å resolution (Fig. 5a; Supplementary Table 2). Clear electron density for all residues from 876 to 937 indicates that they form an extended helix that dimerizes in an antiparallel manner (Fig. 5b; Supplementary Fig. 16). This antiparallel dimerization is stabilized by multiple apolar contacts involving 13 residues from each helix, and six polar interactions involving R892 with D900, and T888 with S906 (via a water molecule) (Fig. 5a; Supplementary Fig. 14B).

At the center of the interface, the structure highlights how residues T888, R892 and V903 contribute to the dimerization (Fig. 5a; Supplementary Fig. 14B). Three mutations (T888D.R892E.V903D) were introduced into the 875–940 peptide to assess the impact on proximal dimerization. Importantly, these residues were chosen on the surface of the 3HB so that the mutations would not disrupt the 3HB stability (Fig. 5c). (Note that residue 892 can be a Gln or an Arg depending on the species (Supplementary Fig. 14B) but both are compatible with the formation of the dimer). SEC-MALS confirms that the T888D.R892E.V903D mutant stays monomeric even up to 43 μM (peak concentration in the SEC-MALS experiment) while the WT counterpart is dimeric in similar conditions (Fig. 5d).

Finally, a model of the active dimeric configuration of Myo6 bound to F-actin including this crystal structure was built (Fig. 5e, see "Methods"). The inter-head distance is indeed compatible with the large (-30 nm) stepping previously reported when Myo6 walks processively[11,41]. Taken together, our results strongly support that proximal dimerization requires the formation of an extended antiparallel coiled-coil, which can form following the destabilization of the 3HB.

## GIPC1 promotes unfolding of the Myo6 monomer and proximal dimerization

We further characterized this proximal dimerization region and investigated the ability of partners to promote proximal dimerization

of Myo6 using an actin-based ATPase assay. Such dimerization indeed leads to "gating", i.e., coordination between the two Heads of the dimer that translates into slowing of ATP binding to the lead Head while the rear Head is attached[42]. This results in a 50% drop in ATPase rate per Head when Myo6 is dimerized compared to a monomer. The ATPase rate of zippered dimer (Myo6 truncated at R991 followed by a leucine zipper to create a constitutive dimer) in which gating has been characterized[42] is indeed -50% that of the monomeric MD<sup>Ins2</sup> ATPase rate (Fig. 5f).

Indeed, upon addition of GIPC1, we found a -50% reduction in the maximal ATPase activity per Head for FLMyo6 (WT) compared to MD<sup>Ins2</sup> (Fig. 5f), consistent with GIPC1 promoting proximal dimerization of FLMyo6. In contrast, the addition of GIPC1 to the FLMyo6 (SAHmimic) mutant is similar to that measured with MD<sup>Ins2</sup>, consistent with a role of GIPC1 in fully freeing the Motor domain from Tail inhibition upon stabilizing an extended, monomeric conformation. Importantly, the FLMyo6 (T888D.Q892E.V903D) exhibits -2-fold higher maximal ATPase rate upon GIPC1 addition, consistent with loss of gating (Fig. 5f).

This additional evidence strongly validates the role of these residues in antiparallel proximal dimerization and the role of this region in controlling motor mechanical properties. Furthermore, we demonstrate that proximal dimerization (involving 3HB unfolding) can be triggered upon GIPC1 binding.

## The L926Q deafness mutation indirectly impairs proximal dimerization

Interestingly, when we used GIPC1 to activate the FLMyo6 (L926Q) construct (Fig. 5f), the maximal ATPase activity that we found was intermediate between monomeric and dimeric FLMyo6. Since our proximal dimerization structure indicates that the L926Q missense mutation does not impact the antiparallel coiled-coil region itself (Supplementary Fig. 15B), and since we found that 3HB unfolding is essential for proximal dimerization, we hypothesized that L926Q impairs Myo6 dimerization by perturbing the unfolding of the 3HB. This was previously reported for the FLMyo6 (SAHmimic) mutant[16].

To monitor 3HB unfolding, we introduced cysteines at two positions of the 3HB surface (T845 and A880), for tetramethylrhodamine (TMR) labeling (Fig. 5c, Supplementary Fig. 15A). As previously described[17], a low TMR fluorescence ratio is found when 3HB is folded (fluorescence quenching due to stacking of the rhodamine rings; MD<sup>Ins2/IQ/3HB</sup> T845C, A880C). This value increases upon 3HB opening (Supplementary Fig. 15A), and a high TMR fluorescence ratio indicative of 3HB unfolding has been reported for the Myo6 zippered dimer T845C, A880C[17]. Introducing the L926Q mutation in this zippered construct led to an intermediate fluorescence intensity, indicating limited unfolding of the 3HB for the deafness mutant compared to control (Table 2). This suggests a role for the L926Q mutation in limiting the conformational changes of the 3HB required for dimerization, in addition to its effect in destabilizing the off-state.

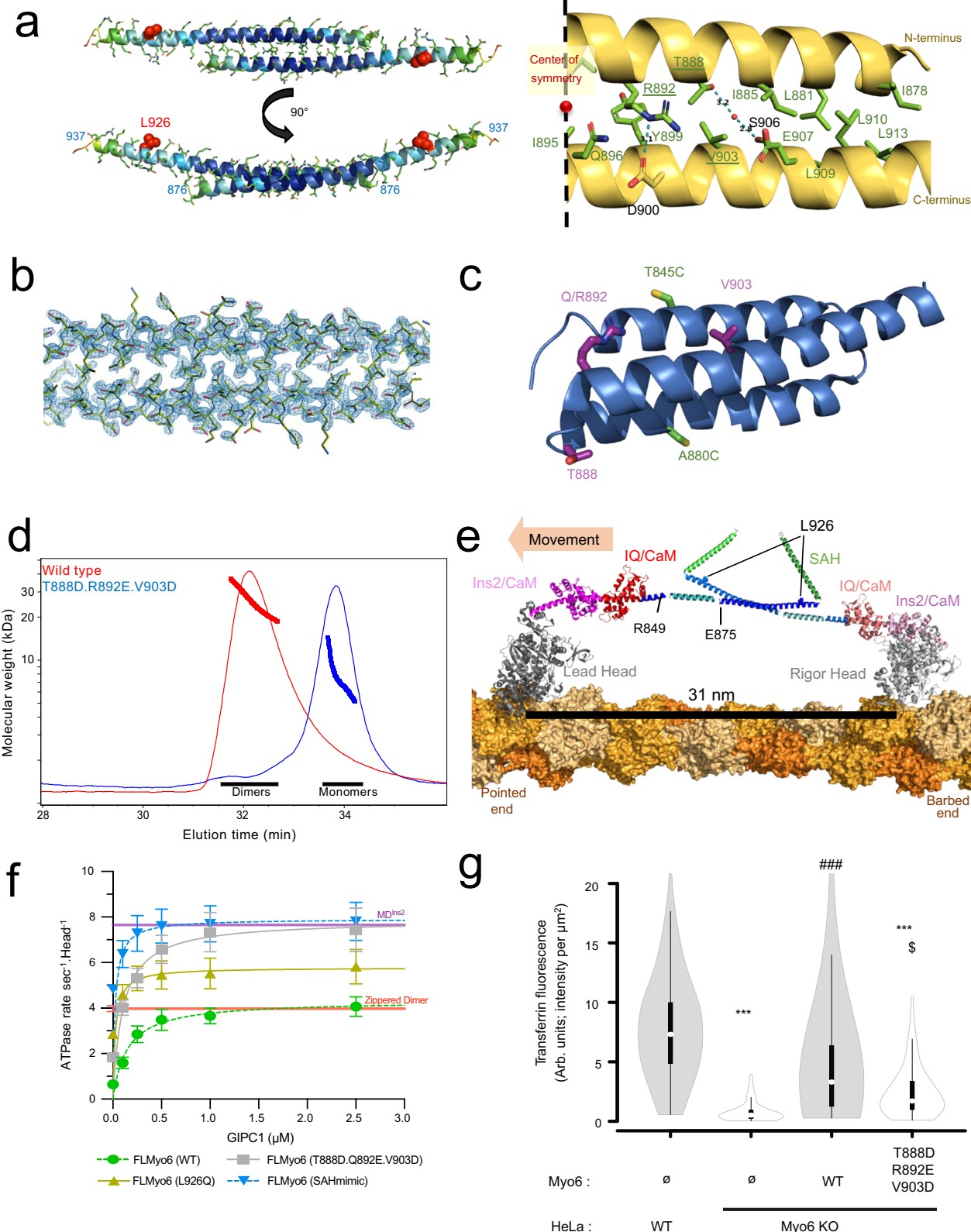

## Importance of proximal dimerization in cells

To further demonstrate that proximal dimerization of Myo6 occurs via the antiparallel coiled-coil seen in our structure, we compared the ability of FLMyo6 (WT) and FLMyo6 (T888D.R892E.V903D) to rescue Myo6-mediated transferrin uptake[16] in HeLa cells whose Myo6 was inactivated using CRISPR/Cas9 (Supplementary Fig. 17A). FLMyo6 (WT) and the FLMyo6 (T888D.R892E.V903D) were

transiently expressed, and the transferrin internalized during a 10 min pulse was quantified. As summarized in Fig. 5g and Supplementary Fig. 17B, expression of the T888D.R892E.V903D mutant, unable to form the proximal dimer, profoundly decreases the rate of uptake of endocytic vesicles, providing evidence for the need for proximal dimerization to optimize Myo6 function during endocytosis. Furthermore, this also strengthens the evidence that

**Fig. 5 | Myo6 can form an antiparallel dimer through residues 875–940 which allow large steps. a** (Left) X-ray structure of mouse Myo6 875–940 antiparallel dimer colored according to B-factor from 18.6 Å$^2$ (dark blue) to 150.8 Å$^2$ (red). (Right) Key residues for dimer stabilization. Apolar contacts are mediated by residues pictured in green. Dotted blue line: polar contacts. Residues mutated in our triple mutant (T888D.R892E.V903D) are underlined. **b** Close-up of the dimerization interface of Myo6 875–940 in the electronic density. **c** Triple helix bundle (PDB: 2LD3) domain. T888, R892, V903, T845C and A880C pictured as sticks are surface residues. **d** SEC-MALS profiles of Myo6 875–940 WT (red) and T888D.R892E.V903D mutant (blue), following injection of 50 µL at 10 mg/mL in 10 mM Tris-HCl pH 7.5; 50 mM NaCl; 5 mM NaN$_3$; 0.5 mM TCEP. Thin lines: static light scattering; thick lines: measured molecular mass. WT elutes as dimers (32 µM concentration at the peak, measured by the in-line refractometer) and T888D.R892E.V903D mutant elutes as monomers (43 µM at the peak). **e** Model of active FLMyo6 dimer (see "Methods"). **f** ATPase rates (mean ± SD) of FLMyo6 WT (green), T888D.Q892E.V903D (gray),

SAHmimic (blue) and L926Q (yellow) at 40 µM F-actin and increasing concentrations of GIPC1 (n = 6). ATPase rates of MD$^{Ins2}$ and zippered dimer[11] without partner (n = 6) plotted as purple and red thick lines (respectively) as references for monomeric and dimeric Myo6. **g** Fluorescence intensity of internalized transferrin was measured for each condition after treatment with genistein (cells examined over 2 independent experiments: WT = 62, KO = 58, KO+ FLMyo6 (WT) = 79, KO+ FLMyo6 (T888D.R892E.V903D) = 66) (p < 0.001, one-way ANOVA; Tukey post-hoc comparisons; one-sided). P-values are the following: WT vs KO: p < 0.0001, WT vs KO+ FLMyo6 (WT): p = 0.3363, WT vs KO+ FLMyo6 (T888D.R892E.V903D): p < 0.0001 (***), KO vs KO+ FLMyo6 (WT): p = 0.0001, KO vs KO+ FLMyo6 (T888D.R892E.V903D): p = 0.5228, KO+ FLMyo6 (WT) vs KO+ FLMyo6 (T888D.R892E.V903D): p = 0.0146 ($).Whisker boxes (10–90 percentile with 2nd and 3rd quartiles within the box; white dot indicates the median) encased within a violin plot (generated with BoxPlotR[70]). (**d**, **f**, **g**) Source data are provided as a Source Data file.

## Table 2 | L926Q stabilizes the 3HB

| Construct | Fluorescence ratio without actin + ATP | Fluorescence ratio with actin + ATP | Molar ratio of labeling per myosin head |
|---|---|---|---|
| MD$^{Ins2/IQ/3HB}$ T845C | 256.1 ± 24.4 | 243.8 ± 14.5 | 1.03 |
| MD$^{Ins2/IQ/3HB}$ T845C, A880C | 22.5 ± 5.8 | 22.6 ± 7.2 | 2.10 |
| Zippered dimers A880C | 238.2 ± 23.7 | 232.3 ± 30.4 | 1.11 |
| Zippered dimers T845C, A880C | 206.5 ± 19.6 | 214.3 ± 12.6 | 2.22 |
| Zippered dimers T845C, A880C, L926Q | 147.53 ± 30.4 | 164.6 ± 18.7 | 2.14 |

Fluorescence observed by TMR labeling of one or two cysteine residues inserted into the three-helix bundle of monomers (MD$^{Ins2/IQ/3HB}$) and zippered dimers. Fluorescence was analyzed by a ratio of the emission values to that of the absorption values for each construct from four independent measurements (n = 4). Mean values (±SD) are reported. The molar ratio was calculated by comparing the myosin concentration to the concentration of the incorporated TMR.

proximal dimerization occurs via an antiparallel coiled-coil as depicted in our crystal structure.

## Discussion

Despite the significance of controlling where and when myosin motors generate forces and move cargoes in cells, careful investigation of how the function of myosin motors is regulated has only been performed for a few classes of myosin[2,3,43,44], and most extensively for Myo2. The results of this study highlight the importance of regulated inhibition of the Myo6 motor until it reaches its target in a cell and is activated. Myo6 must cross actin-rich regions in order to diffuse and reach its binding partners which selectively activate motor activity (Fig. 6a). If the motor was not blocked from interacting with and cycling on actin, Myo6 would bind to actin filaments throughout the cell, retarding diffusion to its target sites at the cell membrane. The fact that the L926Q mutant disrupts the folding and regulation of Myo6 (Fig. 2) and causes deafness in humans[39] attests to the critical need for the regulation of this class of myosin motors.

Our structural and functional studies provide a more precise model to account for the interactions stabilizing Myo6 back-folding (Fig. 1h, Table 1). Among the major differences compared to previous models[8,36], we show that (1) ADP.P$_i$ bound to the Motor domain is essential to lock Myo6 in its back-folded state (Fig. 1b, Table 1); (2) back-folding involves a specific loop of the CBD$^c$ (Table 1), which was previously predicted to be external to the folded complex by Alphafold[45] (Supplementary Fig. 18); and (3) the 3HB/SAH junction acts as a critical hinge to control the equilibrium between on/off states of the motor (Fig. 1g, h, Fig. 2). Earlier studies of the folded monomers[27–29,36] focused only on interactions within a full-length construct in the absence of nucleotide or within a Motor-less construct, and thus did not fully represent what is happening with the full-length Myo6 monomer saturated with nucleotide.

Intriguingly, a single amino acid change (L926Q) causes deafness[39] and is in fact sufficient to destabilize the back-folded

monomer (Fig. 2). This SAH mutation flanks the hinge region that we identified as essential for the off-state of this motor. To further investigate the impact of Myo6 back-folding in myosin recruitment, we used the FLMyo6 (L926Q) and FLMyo6 (SAHmimic) mutants to probe their impact on Myo6 recruitment on melanosomes. What was observed (Supplementary Fig. 8B, C) was that both constructs led to greater recruitment than the FLMyo6 (WT). This is not a gain of function, but rather a loss of regulation as, (1) the normal cellular control over the spatial and temporal recruitment of Myo6 has been lost, and (2) fluorescence quenching assays show that both SAHmimic[16] and L926Q (Table 2) mutations impair proximal dimerization and thus Myo6 function. Deafness due to the L926Q mutation in humans may therefore result from the inability of Myo6 monomers to reach their target sites in hair cells due to loss of folded regulation.

We next examined the ability of some of the Myo6-binding partners that recognize different regions of the CBD to induce unfolding and recruitment of Myo6. Folding not only prevents cycling of the motor on actin until the cellular target has been reached but as shown by our actin-activated ATPase (Fig. 3d), pull-down (Fig. 3e) and recruitment assays on melanosomes (Fig. 4), the folding can also prevent interaction with a subset of cellular partners until unfolding occurs, by either a different class of partners or potentially by a spike in cellular Ca$^{2+}$ concentration[33,36] or PIP$_2$ recognition[46]. We thus propose a model of the folded off-state of FLMyo6 in which the GIPC1 binding site is available for binding, while the TOM1/Dab2 site is masked (Fig. 3a–c). Interestingly, this demonstrates that not all partners are equivalent in their potential for binding the auto-inhibited form of the motor and to activate Myo6.

While TOM1 and Dab2 cannot trigger Myo6 initial unfolding, once bound they prevent the formation of the off-state due to their incompatibility with it, as previously proposed[27,28]. Depending on the nature of the partner and its distribution, the binding will activate the Myo6 motor and could drive either proximal dimerization (Fig. 5f),

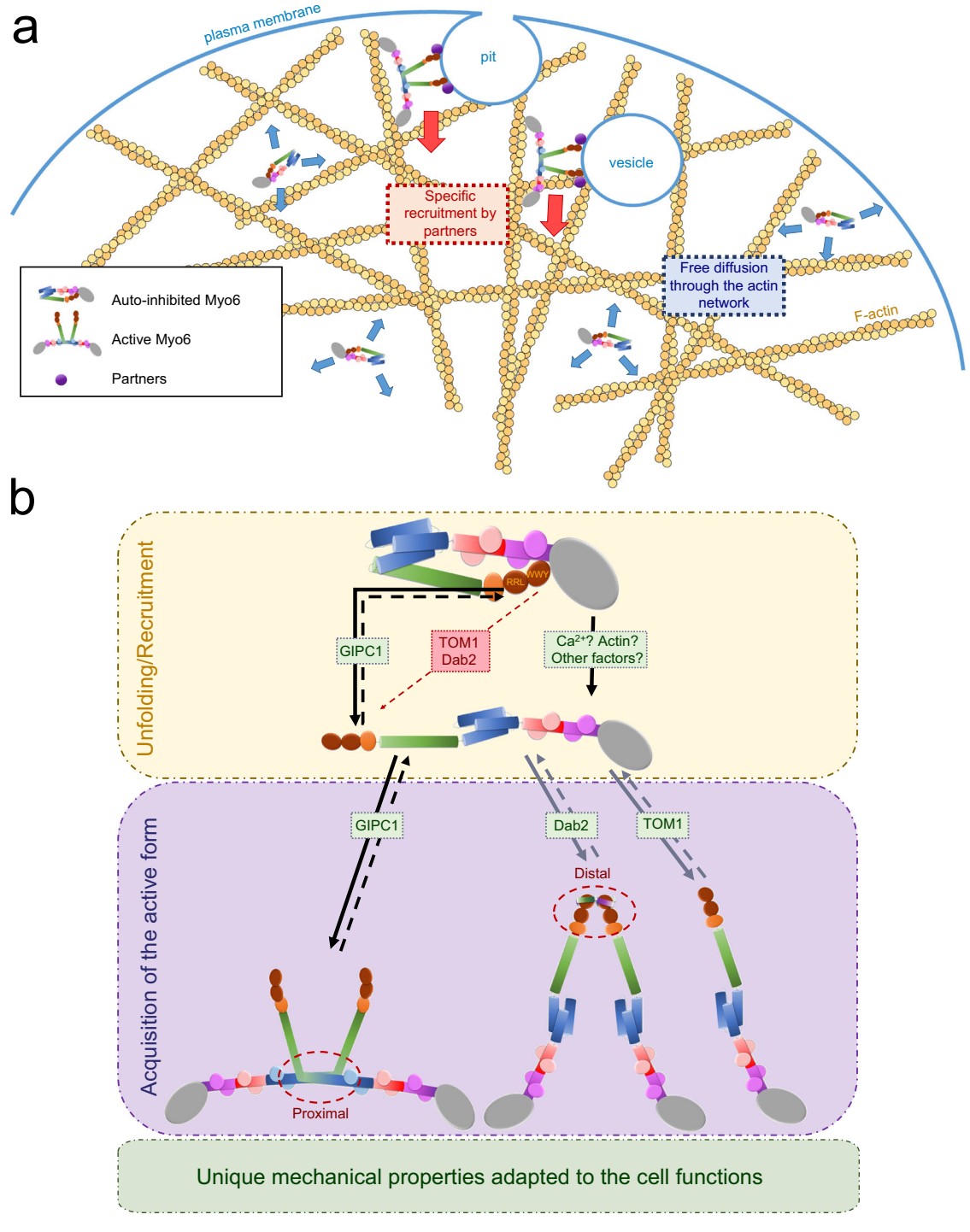

**Fig. 6 | Importance of a folded monomer for regulation. a** When auto-inhibited, Myo6 can diffuse across actin-rich regions and interacts weakly with F-actin. These weak actin interactions (~7 μM apparent affinity, estimated in Supplementary Fig. 4E) result in facilitated diffusion and in increasing the Myo6 concentration in actin-rich regions of the cell. Once recruited by a partner, Myo6 is activated and starts performing its cellular function. **b** Scheme representing possible activation mechanisms for Myo6. Myo6 domains are color-coded: Myo6 MD (gray), Ins2/CaM (purple), IQ/CaM (red/pink), 3HB in blue, SAH (green), DT (orange), CBD (brown), and the partner binding sites (garnet). The binding site

(WWY) for Dab2 and TOM1 is blocked, preventing recruitment of Myo6 without a prior unfolding signal prior to unblock their binding. GIPC1 can bind the accessible RRL motif resulting in Myo6 recruitment and opening. Other signals can act as unfolding factors such as $Ca^{2+}$, which can allow TOM1 to bind to Myo6. Such an activation cascade was previously proposed[36]. Once unfolded, Myo6 potentially acts as a monomer, as previously proposed[35] upon TOM1 binding; or it can dimerize[29] through proximal dimerization, as demonstrated in this study with GIPC1 binding; or it dimerizes through distal dimerization upon Dab2 binding[13], which may lead to proximal dimerization.

distal dimerization[13,14] or maintain an activated, monomeric form[35]. Taken together, these results suggest unique roles for partners not only in Myo6 localization but also in the control of Myo6 activation and function (Fig. 6b).

Once unfolded, binding to its cargo brings two unfolded Myo6 monomers into close apposition, favoring its dimerization[11,13,14,16,17,27]. The experiments summarized in Fig. 5 provide the previously unknown structure of the proximal dimerization region. We present both in vitro

and cellular evidence in support of the structure. This structure reveals the dimerization region to be an antiparallel coiled-coil, as for Myo10[12]. Mutations of three of the amino acids that stabilize this coiled-coil structure (T888D.Q892E.V903D) abolish dimer formation in in vitro assays, but with no impact on back-folding of the monomer (Fig. 5f). Furthermore, the introduction of this Myo6 triple mutant into cells fails to rescue endocytosis (Fig. 5g), providing evidence for the need of proximal dimerization to optimize this cellular function of Myo6.

Myo7A, Myo10 and Myo6 exist as folded monomers in cells until they are activated and recruited by their partners. The formation of an antiparallel dimer may be the mode of dimerization for the three classes that appear to undergo this folded monomer-to-dimer transition. The structure of the active form of Myo6 has been a long-debated issue[11,13–18,33–35], which is resolved by our structure for the proximal dimerization region. As shown in Fig. 5e, Myo6 is unique in that its antiparallel coiled-coil and Lever arm in the dimer are derived from the unfolding of a 3HB, with the contribution of the SAH. The resulting Lever arm formed by the CaM binding region and the unfolded 3HB (half of which contributes to the coiled-coil) is sufficiently long to account for the ability of Myo6 to take steps that average ~30 nm on actin[11,41]. These findings provide a structural framework that can be applied to understanding how motors are recruited and how partners influence motor functions in cells.

## Methods

### Constructs cloning, expression and purification

A list of all the primers and cloning techniques used to clone our constructs can be found in Supplementary Table 3.

### Cloning, expression and purification from Sf9/baculovirus system

The full-length wild-type Myo6 (FLMyo6) was generated using human *MYO6* cDNA splice form without the large insertion (Q9UM54-2 in UNIPROT). The small insert was removed through subcloning to obtain a FLMyo6 construct without any spliced insert (corresponding to iso-form Q9UM54-5). The FLMyo6 (no inserts) construct was then used in all in vitro experiments requiring a full-length construct except for the Anti-His pull-down experiment, for which the small insert isoform was used.

The deafness mutant (L926Q) and triple mutant (T888D.Q892E.V903D-R892 in mouse corresponds to Q892 in human, see Supplementary Fig. 14B) in the antiparallel dimerization region were produced from FLMyo6 with no insert by Quikchange and reverse PCR respectively. The mutant FLMyo6 (SAHmimic)[16] was made where the residues from Glu922 to Glu935 (EAERLRRIQEEMEK) were replaced with alternate acidic and basic residues (EERKRREEEERKK-REEE) to match the (i, i+4) phasing observed in the predicted Myo6 SAH domain.

For microscale thermophoresis and ATPase assays, previously described constructs were used: MD (1-789)[47], MD^Ins2 (2-816)[48], MD^Ins2/IQ/3HB(1-917)[17]. The Myo6 zippered dimer[38] was created by truncation at R991 followed by a leucine zipper.

For the bundle unfolding experiments, a monomeric "cys-lite" construct was made by C-terminal truncation at amino acid Q919 and introduction of C321S, C362S, and C611A. To this construct, either a T845C mutation alone or the combination of T845C and A880C mutations was introduced for rhodamine labeling[17]. A dimeric "cys-lite" construct was made by the introduction of the C321S, C362S, and C611A mutations in the Myo6 zippered dimer[38]. Into this construct, either a T845C mutation alone or the combination of T845C and A880C mutations were introduced for rhodamine labeling[17], with or without the addition of a deafness-causing mutation (L926Q).

Each of these constructs had a Flag tag (GDYKDDDDK) at its N-terminal end to facilitate purification. Expression in baculovirus system and purification were performed as follows:[16]

Sf9 cells were infected with recombinant baculovirus driving high-level expression of our Myo6 construct, and co-infected with recombinant virus containing human CaM. Three days after infection, Sf9 cells were either flash-frozen for later purification or directly used for protein purification. Cells were mechanically lysed by 7 shots in a dounce homogenizer in buffer: 200 mM NaCl; 20 mM HEPES pH 7,5; 4 mM MgCl$_2$; 0.5 mM EDTA; 1 mM EGTA; 0,5% Igepal; 7% Sucrose; 1 mM NaN$_3$; 10 µg/mL Aprotinin; 10 µg/mL Leupeptin; 2 mM DTT; 2 mM ATP. The lysate was centrifuged at 20,000 rpm for 45 min in a 25.50 rotor. The supernatant was incubated with anti-Flag epitope antibody affinity resin under stirring for 2 h at 4 °C. The anti-flag resin was then loaded on a column. The resin was washed with 200 mL of buffer: 150 mM KCl; 20 mM imidazole pH 7,5; 5 mM MgCl$_2$; 1 mM PMSF; 3 mM DTT; 1 mM EDTA; 1 mM EGTA; 10 µg/mL aprotinin; 10 µg/mL leupeptin; 3 mM ATP. The myosin was then eluted via Flag peptide competition. The sample was then ultracentrifuged at 78,000 rpm at 4 °C for 15 min in a TLA110 rotor to spin out any actin that could still be bound to the myosin. The protein was then microdialysized (for ATPase assays) or injected in a Superdex 200 Increase column (Cytiva) and concentrated into the appropriate buffer for the following experiments (with or without nucleotide). The purity of all myosin preparations was confirmed on SDS-PAGE gels.

Jo and In-Flag sequences[37] were synthesized (Eurofins genomics) and fused to Myo6 N-terminus (linker Gly-Ser) and C-terminus (linker Gly), in pVL1392 for expression in Sf9 cells. Purification was achieved using the same protocol as for FLMyo6 except that, for EM studies, purification was performed by replacing ATP with ADP.VO$_4$ in the lysis buffer. For increased purity, a SEC step was performed using a Superdex 200 Increase column (Cytiva) developed in 10 mM Hepes; 80 mM NaCl; 5 mM NaN$_3$; 1 mM MgCl$_2$; 0.1 mM TCEP; 0.1 mM ADP; 0.2 mM VO$_4$; 0.1 mM EGTA; pH 7.5.

### Constructs cloning, expression and purification from *Escherichia coli*

**Cloning Myo6 constructs.** Ins2/IQ/3HB was generated using our human FLMyo6 NI construct (see previous section), DNA sequence encoding for aa 783–917 was transferred into pPROX-HTB plasmid containing in N-terminus 6XHis-tag and a TEV cleavage sequence (coding for ENLYFQG).

Myo6^YFP CBD was generated through several rounds of subcloning from cDNA mouse *MYO6* (E9Q3L1 in UNIPROT). *MYO6* was incorporated in the pET14 plasmid containing an N-terminus 6XHis-tag, a yellow fluorescent protein (YFP) and a TEV cleavage sequence. Finally, *MYO6* was truncated in the N-terminus at the position corresponding to aa M1032 through reverse PCR. The ^YFP CBD (D1157V.Y1159D.D1161R.Q1163V) mutant was generated through point mutations addition using reverse PCR on the ^YFP CBD (WT) construct.

**Cloning partner constructs.** In order to avoid partner (GIPC1, TOM1 and Dab2) degradation and auto-inhibition as previously reported[28,34], we used truncations containing the published Myo6-binding domains[13,22,34,35] instead of FL constructs.

For microscale thermophoresis assays and his-pull-down assays, the ^His GIPC1 construct was generated using cDNA full-length mouse *GIPC1* (UNIPROT Q9Z0G0). *GIPC1* was incorporated in pProEX-HTb plasmid containing N-terminus 6XHis-tag and a TEV cleavage sequence. Finally, *GIPC1* was truncated in the N-terminus at the position corresponding to aa D255 through reverse PCR in order to keep only the GH2 domain, which is sufficient for the interaction with Myo6[34].

For ATPase assays, we used GIPC1 in fusion with the mNeonGreen tag: GIPC1 DNA sequence encoding for residues 238-end was incorporated in the pET28 plasmid containing in the N-terminus 6XHis-tag and a mNeonGreen tag using homemade Gibson Assembly mix.

For microscale thermophoresis and his-pull-down assays, a non-fluorescent [His]TOM1 construct was generated using full-length human *TOM1* (UNIPROT O60784-1) cDNA. TOM1 207-end was incorporated into pET14 in-frame with an N-terminus 6XHis-tag and a TEV cleavage sequence.

For ATPase assays, we used the TOM1 436–461 peptide described as the minimal sequence required for TOM1 binding to Myo6[35]. DNA sequence coding for aa 436–461 was incorporated in pET14 plasmid containing an N-terminus 6XHis-tag, yellow fluorescent protein (YFP) and a TEV cleavage sequence.

Dab2 His-650-end (tDab2[28]) was a kind gift of Christopher Toseland.

To identify the minimal sequence involved in proximal dimerization, several Myo6 truncations were generated from mouse *MYO6* cDNA (UNIPROT: E9Q3L1). Constructs encoding for aa 875–940 and 875–955 were cloned with a N-terminal 6xHis-tag into the pET14 plasmid. The construct encoding for aa 834–955 was cloned into pET14 with an N-terminal 6xHis-tag followed by a thrombin cleavage site (coding for LVPRGSH). Constructs encoding for aa 880–940 and 888–940 were cloned with a C-terminal 6xHis-tag into pET14 by PCR and blunt-end ligation. The 912-end construct was cloned into pProEX-HTb with an N-terminal 6xHis-tag through several rounds of subcloning. For crystallization assays, the construct encoding for aa 875–940 was generated with an N-terminal 6xHis-tag and a TEV cleavage sequence into pProEX-HTb using homemade Gibson Assembly mix. The point mutations T888D, R892E, and V903D were added to the backbone encoding for his-rTEV-875-940.

**Protein expression and purification.** Constructs were expressed in *E. coli* BL21 (DE3) cells (NEB). Cells were grown in 2xYT media until $OD_{560}$ ~ 0.8, expression was then induced by the addition of 200 μM isopropyl β-D-1-thiogalactopyranoside (except for Dab2 expression, where 1 mM isopropyl β-D-1-thiogalactopyranoside was used). Cells were lysed by sonication. For purification, the lysate soluble fraction was loaded on an IMAC column (cOmplete 5 mL, Roche for all constructs except [YFP]CBD (WT and mutant) for which HisTrap-FFcrude 5 mL, Cytiva was used instead), and proteins were eluted with 200 mM or 300 mM Imidazole. Purest fractions were identified by SDS-PAGE. If needed, pooled fractions were concentrated using Vivaspin concentrators (Sartorius) up to ~5 mL. Concentrated samples were injected in Superdex 200 or 75 16/600 columns (Cytiva) depending on the molecular weight of the target protein. Purest fractions and the final sample were concentrated by ultrafiltration, and protein concentration was determined using Nanodrop 2000 (ThermoScientific). The final sample containing concentrated protein was flash-frozen in liquid nitrogen and stored at −80 °C.

For proteins containing a TEV cleavable His-tag, prior to the gel filtration, His-tag was removed by incubation with homemade rTEV protease overnight in a 1/50 mass ratio. The incubate was passed through the cOmplete His-Tag Purification Column again to remove rTEV and the uncleaved fraction, then concentrated and loaded in a Superdex 75-16/60 gel filtration column.

Purification buffers are detailed in Supplementary Table 4.

**Constructs cloning for expression in cells**
For expression in MNT-1, FLMyo6 was generated from cDNA of full-length human *MYO6*, no inserts isoform (UNIPROT: Q9UM54-5) with shRNA resistance. DNA was transferred to the pEGFP-C1 vector via an XbaI restriction enzyme site. SAHmimic mutations (Glu922 to Glu935) (EAERLRRIQEEMEK) replaced with alternate acidic and basic residues (EERKRREEEERKKREEE) were introduced by reverse PCR. The L926Q mutation was introduced using Quikchange. Transfer of Jo-Myo6-In from baculovirus vector to P-EGFP-C1 was ordered from GenScript. Myo6 CBD was generated by transferring DNA encoding G1037-end

from human *MYO6*, no inserts isoform (UNIPROT: Q9UM54-5) into the pEGFP-C1 plasmid using the XbaI restriction enzyme site.

I1072A was introduced in previously cloned constructs (see above) using reverse PCR.

Mouse GIPC1 (239-end), human TOM1 (299-end as described in ref. 22) and human Dab2 (650-end) were transferred in a modified pmCherry-C1 plasmid containing in N-terminus a melanosome-targeting tag (MST tag, aa 1–139 from Mouse *MREG*–UNIPROT: Q6NVG5) as described in ref. 40. The MST tag and mCherry are separated by a GGSGGTGG linker. In the [mCherry]MST-partners constructs, mCherry and GIPC1, TOM1 or Dab2 sequences are separated by the polylinker multiple cloning site SGLRSRAQASNSLTSK.

For expression in HeLa cells, our previously existing [Flag]FLMyo6(WT) (pig/mouse) with small insert[11] was introduced in TREX Pcdna4/TO plasmid together with a C-terminal mApple for detection. Mutations (T888D, R892E and V903E) were successively introduced in this construct.

**SEC-SAXS**
SAXS data were collected on the SWING beamline at synchrotron SOLEIL (France)[49] in HPLC mode at λ = 1.0332150494700432 Å using a Dectris EIGER-4M detector at a 2 m distance. Protein samples were injected at 0.1 mL/min on Superdex 3.2/300 column pre-equilibrated in 20 mM HEPES; 200 mM NaCl; 2 mM $MgCl_2$; 1 mM NaADP; 1 mM $AlF_4$, 0.1 mM EGTA, 1 mM DTT; or 20 mM HEPES; 400 mM NaCl; 2 mM $MgCl_2$, 0.1 mM EGTA, 1 mM DTT; pH 7.5 prior to data acquisition in the SAXS capillary cell. Then, 150 frames of buffer scattering (before the void volume) and then 719 frames of elution sample scattering were collected. Exposure time was 1990 ms/frame. Images were processed using the Foxtrot 3.5.10-3979[49] developed at the SOLEIL synchrotron: buffer averaging, buffer subtraction from the corresponding frames at the elution peak, and sample averaging were performed automatically. Further data analysis to obtain Rg, I(0), Dmax and molecular weight estimation was done with PRIMUS from ATSAS suite[50]. Dimensionless Kratky plot $((q.Rg)^2.I(q)/I(0)$ versus $q.Rg)$ was generated using Microsoft Excel based on I(0) and Rg values found with PRIMUS. Twenty envelopes were generated independently with GASBOR[51] and averaged with DAMAVER[52].

**SEC-MALS**
For SEC-MALS analysis, samples were injected in a Superdex 200 10/300 Increase (Cytiva) previously equilibrated in the corresponding buffer, and developed at 0.5 mL/min. Data collection was performed every 0.5 s with a Treos static light scattering detector, and a t-Rex refractometer (both from Wyatt Technologies). The concentration and molecular mass of each data point were calculated with the software Astra 6.1.7 (Wyatt Technologies).

**Microscale thermophoresis measurements between Myo6 tail and head**
Microscale thermophoresis experiments were performed on a Monolith NT.115 system (NanoTemper Technologies) using YFP-fusion proteins.

The non-fluorescent protein was first treated with 0.5 mM EGTA (±2 mM MgADP; 2 mM $Na_3VO_4$ for some experiments); then dialyzed against 20 mM Hepes pH 7.5, 50 mM NaCl, 2.5 mM $MgCl_2$, 1 mM TCEP and 0.05% (v/v) Tween-20 (±2 mM MgADP; 2 mM $Na_3VO_4$ for some experiments). Two-fold dilution series (16 in total) of the non-fluorescent protein (Head sample) was performed at 25 °C in the same buffer. The YFP-fused partner was kept at a constant concentration of 100 nM. The samples were loaded into premium capillaries (Nanotemper Technologies) and heated for 30 s at 60% laser power. All experimental points were measured twice. The affinity was quantified by analyzing the change in thermophoresis as a function of

the concentration of the titrated protein using the NTAnalysis software provided by the manufacturer.

## Microscale thermophoresis measurements with Myo6 partners

Two-fold dilution series (16 in total) of the non-fluorescent protein (Myo6 partner) were performed at 25 °C in the MST buffer: 20 mM Bis-Tris pH 6.5, 100 mM KCl, 1 mM DTT and 0.05% (v/v) Tween-20. The YFP-fused partner was kept at a constant concentration of 100 nM. Microscale thermophoresis experiments were then performed in similar conditions as above in 20 mM Bis-Tris pH 6.5, 100 mM KCl, 1 mM DTT and 0.05% (v/v) Tween-20. Capillaries were heated for 30 s at 50% laser power.

## Protein cross-linking and mass spectrometry detection

Before the cross-linking reactions, and to prevent cross-linking reactions with MgATP/ADP, concentrated full-length myosin 6 was buffer exchanged by 20-fold dilution and concentration (twice) using EDTA buffer (10 mM HEPES, 50 mM NaCl, 2 mM EDTA, 0.1 mM DTT, pH 7.4) to strip the nucleotide from the motor domain. Then, the protein was diluted and concentrated again using Mg buffer (10 mM HEPES, 50 mM NaCl, 2 mM MgCl2, 0.1 mM DTT, pH 7.4) until a concentration of ~4–5 mg/mL. Finally, 0.5 mM 2-chloroadenosine-5′-triphosphate (Cl-ATP) was added to the protein solution.

Protein cross-linking reactions were carried out at room temperature for 60 min at a 1:300 protein to DSSO ratio and quenched with the addition of Tris buffer to a final concentration of 50 mM. The quenched solution was reduced with 5 mM DTT and alkylated with 20 mM iodoacetamide. The SP3 protocol as described in refs. [53,54] was used to clean up and buffer exchange the reduced and alkylated protein. Shortly, proteins are washed with ethanol using magnetic beads for protein capture and binding. The proteins were resuspended in 100 mM $NH_4HCO_3$ and were digested with trypsin (Promega, UK) at an enzyme-to-substrate ratio of 1:20, and protease max 0.1% (Promega, UK). Digestion was carried out overnight at 37 °C. Clean-up of peptide digests was carried out with HyperSep SpinTip P-20 (ThermoScientific, USA) C18 columns, using 80% Acetonitrile as the elution solvent. Peptides were then evaporated to dryness via Speed Vac Plus. Dried peptides were suspended in 3% Acetonitrile and 0.1% formic acid and analyzed by nano-scale capillary LC-MS/MS using an Ultimate U3000 HPLC (Thermo-Scientific, USA) to deliver a flow of approximately 250 nl/min. Peptides were trapped on a C18 Acclaim PepMap100 5 μm, 100 μm × 20 mm nanoViper (ThermoScientific, USA) before separation on PepMap RSLC C18, 2 μm, 100 A, 75 μm × 50 cm EasySpray column (ThermoScientific, USA). Peptides were eluted on a 90 min gradient with acetonitrile an interfaced via an EasySpray ionization source to a quadrupole Orbitrap mass spectrometer (Q-Exactive HFX, ThermoScientific, USA). MS data were acquired in data-dependent mode with a Top-25 method, high-resolution scans full mass scans were carried out (R = 120,000, m/z 350–1750) followed by higher energy collision dissociation (HCD) with stepped collision energy range 21, 27, 33% normalized collision energy. The tandem mass spectra were recorded (R = 30,000, isolation window m/z 1.6, dynamic exclusion 50 s). Cross-linking data analysis: Xcalibur raw files were converted to MGF files using ProteoWizard[55] and cross-links were analyzed by MeroX[56]. Searches were performed against a database containing known proteins within the complex to minimize analysis time with a decoy database based on peptide sequence shuffling/reversing. Search conditions used 3 maximum missed cleavages with a minimum peptide length of 5, cross-linking targeted residues were K, S, T, and Y, cross-linking modification masses were 54.01056 Da. Variable modifications were carbmidomethylation of cysteine (57.02146 Da) and Methionine oxidation (15.99491 Da). The false discovery rate was set to 1%, and assigned cross-linked spectra were manually inspected.

## ATPase assays

Steady-state ATPase activities were measured at 25 °C using an NADH-coupled assay[38]. ATPase rate determined from 2-3 preps with 2-3 independent assays per prep. Myo6 was used at 150 nM, F-Actin was used at 40 μM (unless otherwise noted) and 2.5 μM additional CaM was added in all our experiments. The experiments were all carried out in 10 mM MOPS pH 7.0; 10 mM KCl; 1 mM DTT; 1 mM $MgCl_2$; 1 mM EGTA.

## Anti-His pull-down assay

FLMyo6 SI (WT) or (SAHmimic) were used alone or mixed with partner GIPC1 (His-rTEV-GIPC1 255-end) or TOM1 (His-TOM1 207-492) in a ratio (1/1) (10 μM) and 1 μM of extra Calmodulin was added in a total volume of 20 μL. The input was incubated with 40 μL of $Ni^{2+}$ beads from cOmplete column (Roche), which were previously equilibrated either in ADP.$VO_4$ Buffer (10 mM HEPES pH 7.5; 100 mM NaCl; 5 mM $NaN_3$; 1 mM $MgCl_2$; 0.1 mM TCEP; 1 mM NaADP; 1 mM $Na_3VO_4$; 0.1 mM EGTA; 4 mM imidazole) or ADP.$VO_4$-$CaCl_2$ Buffer (10 mM HEPES pH 7.5; 100 mM NaCl; 5 mM $NaN_3$, 1 mM $MgCl_2$, 0.1 mM TCEP, 1 mM NaADP; 1 mM $Na_3VO_4$; 4 mM imidazole; 1 mM $CaCl_2$) or NF Buffer (10 mM HEPES pH 7.5; 100 mM NaCl; 5 mM $NaN_3$; 1 mM $MgCl_2$; 0.1 mM TCEP; 0.1 mM EGTA; 4 mM imidazole). All steps were performed at 4 °C. Beads were washed by centrifugation after 1 h of gentle agitation. Bound proteins were eluted in 600 mM imidazole in the corresponding buffer.

## Electron microscopy

Purified Jo-Myo6-In at 50 μg/mL in 10 mM HEPES; 80 mM NaCl; 1 mM $MgCl_2$; 0.1 mM TCEP; 0.1 mM ADP; 0.2 mM $Na_3VO_4$; 0.1 mM EGTA, pH 7.5 was transferred to Carbon Film 300 mesh copper grids (Electron Microscopy Sciences), then stained with 2% uranyl acetate. A total of 284 images were collected with a 200 kV Tecnai G2 microscope under low dose conditions with a 4Kx4K F416 TVIPS camera at 0.213 nm/px and treated with the software CryoSPARC[57]. Following CTF determination, template picking was carried out using an initial set of 100 manually picked particles. The resulting 711,671 particles were submitted to a few rounds of 2D classification from which 93,293 particles were selected. These were used in the ab initio reconstruction that produced the map at 17 Å resolution (FSC).

## Model of the Myo6 off-state

We first positioned the Motor domain-Lever arm (residues 1–917) in the Jo-Myo6-In map from negative staining EM. The best model-to-map agreement was obtained with PDB 4ANJ (Motor domain and insert-2/$Ca^{2+}$-CaM with ADP.$P_i$ analog bound, in pre-powerstroke state (PPS)). The Lever arm from PDB 3GN4 was then superimposed to PDB 4ANJ by using the insert-2/$Ca^{2+}$-CaM region, present in both structures, as reference. In the negative staining reconstruction, Jo-In PDB 5MKC was placed in the distinct density that corresponds to it, as expected, with the N- and C-termini pointing toward the center of the main density body occupied by Myo6. The structure of the C-terminal half of the CBD from PDB 3H8D was placed according to structural and biochemical restrictions as follows: (1) the $CBD^c$ C-terminus must be near the N-terminus of the fusion protein In; (2) residues D1157, Y1159, D1161 and Q1163 are in contact with the $MD^{Ins2}$; (3) there is still density to be filled close to the N-terminus of the $CBD^c$, that can be filled by $CBD^n$. (Note that this proposed position is opposite to that currently predicted by Alphafold[45] for uniprot entry: Q9UM54, due to lack of data for the intermolecular interactions when that model was built). At last, the NMR structure of the SAH domain (residues 919–998; PDB: 6OBI) was accommodated in the density. This density is narrowed up to residue ~955 and then becomes much larger to account for the rest of the model, in which no distinct subdomain can be identified. Thus, our current model lacks the distal Tail (a compact domain of 3 nm in diameter[8]) and $CBD^n$, for which there seems to remain enough density to be fitted. Model and figure were prepared with Pymol[58]. The

complete model of the Myo6 off-state is presented in Fig. 3a. Placement of the different domains is further supported by the crosslink experiment presented in Supplementary Fig. 7 and Supplementary Table 1. Its ability to fit our off-state Myo6 SAXS data was tested (Supplementary Figs. 3D–F, 5B, C).

## MNT-1 cell transfection

MNT-1 cells (human pigmented melanoma cell line kindly provided by Pr. Michael S. Marks (Department of Pathology and Laboratory Medicine, Children's Hospital of Philadelphia Research Institute, Philadelphia, PA; Department of Pathology and Laboratory Medicine and Department of Physiology, Perelman School of Medicine, University of Pennsylvania, Philadelphia, PA)) were cultured in DMEM supplemented with 20% FBS, 10% AIM-V medium, 1% sodium pyruvate, 1% nonessential amino acids, and 1% penicillin-streptomycin. For plasmid transfection, 400,000 MNT-1 cells were transfected using nucleofection (NHEM kit, Lonza) on Amaxa device 2 (program T20) with 1.5 μg of [iRFP]VAMP7 plasmid; 1 μg of [mCherry]MST plasmid and 3 μg of [GFP]FLMyo6 plasmid. After transfection cells were seeded in fluorodish containing 1 mL RPMI medium, then 1 mL of complete MNT-1 medium supplemented by 10% FBS was added 6 h post-transfection. The medium was changed 1 day post-transfection by complete medium then cells were fixed with 4% PFA at 48 h post-transfection. Cells were stored in the dark at 4 °C in PBS medium until imaging. The fluorescence intensity of each [mCherry]MST construct was analyzed to ensure equivalent expression levels between the different partners (Supplementary Fig. 19).

## Super resolution imaging and analysis

Samples were imaged in fluorodish using a 100x/1.4 NA oil immersion objective on an inverted Spinning disk confocal microscope (Inverted Eclipse Ti-E Nikon, Spinning disk CSU-X1, Yokogawa) equipped with a Photometrics sCMOS Prime 95B Camera (1200×1200 pixels). Z images series were acquired every 0.2 μm. Images were processed with a Live super Resolution module (Live-SR; Gataca systems) based on structured illumination with optical reassignment technique and online processing leading to a two-time resolution improvement[59]. For the figures, Z maximum projection and a substract background (50 pixels) were applied on SR images using the FIJI software. Analysis was done on raw SR images. Melanin pigments (black spots) were automatically detected in a defined region of interest (ROI) (here, cell outlines that were manually drawn) in BrightField images by creating a MIN-intensity z-projection and considering the lowest values, defined using the 'Find Maxima' function of ImageJ/Fiji and whose spatial coordinates were recorded. To quantify the percentage of melanosomes containing [iRFP]VAMP7/[mCherry]MST/[GFP]Myo6 proteins at the membrane, additional ROIs centered around each individual detected pigment were generated whose size was defined (0.350 μm diameter). Then, for each detected brightfield spot, an additional automatic detection in the fluorescent channel(s) of interest was performed by creating a MAX intensity z-projection in the ROI around the pigments and considering the highest values. Detected pigments were considered positive for the marker of interest above a threshold (defined by Triangle's automatic thresholding method, calculated on the MAX intensity projection, or manual thresholding in the case of cells expressing the lowest GFP-Myo6), and the percentage of which was calculated. Pigments that were automatically detected very close to each other (within 4 pixels in XY and 2 pixels in Z) and that had overlapping ROIs were automatically removed and eliminated from the analysis to avoid data duplication. Moreover, automatically detected pigmented that were negative for [iRFP]VAMP7 and/or [mCherry]MST fluorescent signal were excluded from the analysis because not considered as pigmented melanosomes (positive for membrane-associated components). For each cell, a percentage of Myo6-positive melanosome was calculated and normalized to the total number

pigmented melanosomes (co-positive for pigment and [iRFP]VAMP7 and/or [mCherry]MST).

## Proximal dimer crystallization, data collection, and structure determination

The Myo6 875–940 construct was crystallized by hanging drop vapor diffusion at 290 K by mixing 1 μL of 9.8 mg/mL protein solution with 1 μL of reservoir solution (27% PEG 4000, 10 mM MgCl$_2$ and 0.2 M imidazole/malate, pH 6.0). Crystals grew spontaneously as rods 1 to 7 days after. After an additional 3 weeks, they were cryo-cooled in liquid nitrogen in a solution containing 28% PEG 4000, 10 mM MgCl$_2$, 0.2 M imidazole/malate, pH 6.0, and 27% ethylene glycol. One exploitable X-ray dataset was collected at the Proxima 1 beamline (Synchrotron Soleil, Gif-Sur-Yvette) and processed with Autoproc[60]. Diffraction limits after treatment with Staraniso[61] with a cutoff of 1.2 I/sI were 2.566 Å in two directions, and 2.077 Å[61] in one direction. Initial structure factors were obtained by molecular replacement with Phaser[62] using a helix comprised of 30 serine residues as search model. Initial sequence attribution was obtained with Phenix AutoBuild[63], followed by several cycles of iterative edition with Coot[64] and refinement with Buster[65]. Resolution was automatically cut by Buster to 2.2 Å based on model-map cross-correlation. The dimer is defined by one of the 2-fold symmetry axis, with crystal contacts between the N-terminus of one dimer and the C-terminus of neighboring dimers (Supplementary Fig. 16B, C, Supplementary Movie 2). When the carbons are colored according to B-factors (Fig. 5a), the lowest values are found between residues 885 and 913, suggesting that the dimerization interface is comprised within those boundaries.

## Model of the Myo6 proximal dimer

The cryo-EM structure of Myo6 bound to actin (PDB: 6BNP) was used as basis for placing two Myo6 Motor domains in rigor and PPS states (gray) at a distance compatible with the average step size of ~30 nm previously measured for the truncated constructs (at 991 or 1050), the zippered dimer and the full-length protein[11] (Fig. 5e). On each side, the N-terminus of the Lever arm bound to two light chains (pink, PDB: 3GN4) was aligned to the corresponding residues in the Converter. The crystallized dimerization domain (blue) was then placed at a minimal distance from the two Lever arms, leaving a gap of 4.6 nm from each side. This gap needs to be filled by a stretch of 26 amino acid residues that would make 3.9 nm if in a theoretical helix, or up to 9.9 nm if fully extended. SAH (green) (PDB: 6OBI) was connected to the C-terminus of the dimerization region via a putative kink. Model and figure were prepared with Pymol[58].

## Bundle unfolding assay

As previously described[17], cysteine residues were introduced to replace T845 and A880 in Myo6-917 (MD[Ins2/IQ/3HB]) and Myo6-991-leucine zipper (zippered dimer) constructs with no reactive cysteines. Control constructs contained one reactive cysteine, T845C. One mg of each protein was labeled with a 10-fold molar excess of TMR 5-iodoacetamide (5-TMRIA; Anaspec, San Jose, CA) per cysteine (from a stock concentration of 20 mM in dimethylformamide) at 4 °C for 1–3 h. Unbound rhodamine was removed by gel filtration and overnight dialysis. Absorption spectra were measured in an HP Diode Array Spectrophotometer, and fluorescence spectra were obtained in a PTI QM3 luminescence spectrofluorometer. The excitation and emission spectra were measured at 552 and 575 nm, respectively.

## Generation of Myo6 null HeLa cells

The *MYO6* gene of HeLa cells (ATCC CCL-2) was inactivated by the CRISPR/Cas9 gene editing approach. Briefly, HeLa cells were transfected using a combination of three human Myo6 CRISPR plasmid variants (Santa Cruz Biotech SC-401815)—each driving expression of Cas9, GFP and one of the following human Myo6-specific 20-

nucleotide gRNAs (5′–3′: taatatcaaagttcgatata, acattctgattgcagtgaatc, ccaagtgtttcctgcagaag). Clones of transfected HeLa cells were selected on the basis of GFP fluorescence, and PCR of isolated DNA using primers flanking the targeted genomic sequences. Loss of Myo6 expression was confirmed by western blot using an anti-Myo6 antibody (rb pc)(EMD-Millipore, Cat# ABT42, Lot# 3011368, used at 1:1000 dilution for the western blot).

## Transferrin endocytosis assay

Normal and Myo6 null HeLa cells were grown in multi-well tissue culture plates on coverslips coated with rat collagen I (Corning). FLMyo6 (WT) or FLMyo6 (T888D.Q892E.V903D) were tagged with a C-terminal mApple for the identification of expressing cells. Transfections were performed using the X-tremeGENE 9 DNA transfection reagent (Sigma-Aldrich) following the manufacturer's instructions. Cells were serum starved, but otherwise maintained in normal growth conditions−at 37 °C with 5% $CO_2$, by incubation in serum-free Dulbecco's modified Eagle's medium for 2.5 h. Serum-free medium was supplemented with genistein (600 μM, Cayman Chemical Company) to inhibit caveolae-mediated uptake of transferrin following Myo6 depletion as previously reported[16]. During the final 10 min of serum starvation, Alexa fluor 488-conjugated transferrin (ThermoFisher) was added to the culture medium at 25 μg/mL. Following serum starvation, plates were placed on ice and washed twice with 10 mM HCl and 150 mM NaCl to remove cell surface-bound transferrin. The cells were fixed with ice-cold 4% paraformaldehyde for 20 min and stained with rabbit anti-dsRed antibody (rb pc) (TakaraBio, Cat# 632496, Lot# 2103116, used at 1:4000 dilution for IF) and Alexa fluor 568-conjugated anti-rabbit secondary antibody to identify cells expressing mApple-tagged Myo6. Image acquisition was performed with Leica Application Suite X software on the Leica TSC-8 confocal system using a 40X oil immersion objective lens (n.a. = 1.3). Transferrin uptake was determined using ImageJ software: the total transferrin-conjugated fluorescence intensity from sum slice projections of individual cells was subsequently normalized by cell size. Comparative samples were stained, imaged, and processed simultaneously under identical conditions. Data were subjected to a one-way analysis of variance with Tukey post-hoc comparison of individual groups to determine statistical significance.

## Statistics and reproducibility

For the transferrin endocytosis assay, cells were examined over 2 independent experiments (number of cells examined by conditions: WT = 62, KO = 58, KO+ FLMyo6 (WT) = 79, KO+ FLMyo6 (T888D.R892E.V903D) = 66). All fluorescence images were acquired from random fields of view. A one-sided one-way analysis of variance (ANOVA) with Tukey post-hoc comparison was performed. For studying Myo6 recruitment to melanosomes, ~20–30 cells were examined over 2–3 independent experiments for each condition. For statistical analysis, a two-sided unpaired $t$-test with Welch's correction was performed. Only cells for which both Myo6 construct and MST construct transfection were successful were taken into account. Non or less-pigmented cells, cells with low fluorescence intensity and/or low expression precluding signal thresholding and further quantification were excluded (~11% of cells excluded). Statistical analyses were performed with GraphPad Prism. Sample sizes were chosen to reach statistical significance, and data were reproducible. The investigators were not blinded.

## Reporting summary

Further information on research design is available in the Nature Portfolio Reporting Summary linked to this article.

## Data availability

The atomic model of the Myo6 proximal dimer generated in this study is available on the PDB[66] under the accession code 8ARD. The other atomic coordinates used in this study are available on the PDB[66] under the accession numbers 3H8D, 6J56, 5V6E, 2LD3, 4ANJ, 3GN4, 5MKC, 6OBI and 6BNP. The mass spectroscopy data supporting Supplementary Fig. 7 data have been deposited to the ProteomeXchange Consortium via the PRIDE[67] partner repository with the dataset identifier PXD044767. Source data are provided with this paper.

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

## Acknowledgements

The authors greatly acknowledge the Cell and Tissue Imaging (PICT-IBiSA)–Institut Curie, member of the French National Research Infrastructure France-BioImaging (ANR10-INBS-04) and in particular Anne Sophie Macé for her help in the quantification of the results; the CurieCoreTech Mass Spectrometry Proteomics and Recombinant protein; the beamline scientists of PX1 and SWING (SOLEIL synchrotron) for excellent support during data collection; Pierre-Damien Coureux for preliminary experiments in negative staining electron microscopy; Margaret Titus for her help with Jo-Myo6-In cloning and comments on the manuscript; Virginie Ropars for her help in SAXS data optimization, Guillaume Jousset for his advises in biochemistry and Christopher Toseland for providing us with Dab2 DNA. H.L.S. was supported by the National Institutes of Health Grant DC009100. A.H. is supported by grants from CNRS and ANR-19-CE11-0015-02. A.H. and C.D. were supported by grants from ANR-17-CE11-0029-01. The A.H., D.L. and C.D. teams are part of the Labex Cell(n)Scale ANR-11-LBX-0038 and IDEX PSL (ANR-10-IDEX-0001-02-PSL). K.P. is a recipient of a Marie Curie fellowship 797150 MELANCHOR. V.P.-H. and L.C. are recipients of a PhD fellowship from Ligue contre le cancer GB/MA/SC-12630 and IP/SC-16058. J.S.-C. as a recipient of a fellowship from the Fondation pour la Recherche Médicale (SPF201909009097).

## Author contributions

A.H. and H.L.S. designed and directed the research. L.C., V.J.P.-H., K.J.P. and C.D. were involved in project management. H.S., X.L., L.H., L.C., T.L., C.K., L.V., A.D., E.F., P.H., V.J.P.-H. and D.M. cloned, produced and purified constructs. L.C., T.L., C.K., H.S., L.H., A.D., L.V., P.H., V.J.P.-H. and D.M. performed biophysical and biochemical assays. C.K. performed grid optimization for negative staining EM that was collected by A.D.C. D.L. provided access to the 200 kV microscope. E.D. provided access to mass spectrometry equipment. C.K. analyzed the negative staining data and performed the 3D reconstruction. H.S. characterized the limits of the proximal dimerization region, crystallized it and C.K. solved the structure. F.M., Y.I.L. and K.J.P. performed the cell-based assays. T.E.M. performed cross-linking and mass spectrometry sample preparation and analysis. L.C., C.K., A.H., H.L.S., F.M., Y.I.L., H.S., L.H., A.D., L.V., K.J.P., T.E.M. and V.J.P.-H. analyzed the data. J.S.-C. provided technical help in the quantification of the cell-based assays. L.C., A.H., C.D., C.K. and H.L.S. wrote the initial version of the paper, with the help of F.M. and V.J.P.-H. and all authors reviewed it. A.H. and H.L.S. provided funding.

## Competing interests

The authors declare no competing interests.
