## [Peer Review File · Nature Communications]

REVIEWER COMMENTS

Reviewer #1 (Remarks to the Author):

The manuscript by Canon et al provides a support for model of myosin 6 compact off state and how the molecule might dimerize via an antiparallel coiled coil that is formed upon opening of the well described three helix bundle motif in this myosin. The SAX model for the back-folded off state is explained by placing various known crystal structures of myosin 6 fragments and speculating on what part of the myosin structure could fill the remaining unaccounted for spaces in the SAX envelop. Several different methodologies are used to generate and support these models including SAX, protein:protein binding experiments, size exclusion chromatography, enzymatic assays, crystallography and intercellular localization experiments. For the most part the data are well presented and convincing. The results will be of great interest to structural biologists, cell biologists and the communities generally interested in myosin structure and function as well as the deafness. I do have some questions and issues that should be addressed.

- 1) The model shown in in Fig. 6A and in many places in the text assumes that the back-folded conformation of myosin 6 does not bind to actin with high affinity. The covalently stabilized Jo-myosin 6-In construct seems to be an ideal fragment to use in actin sedimentation experiments to quantify this affinity. Could such an experiment be provided?
- 2) Line 133: Show a field of negatively-stained FLMyo6 in ADP.VO4 in the supplementary figures for comparison.
- 3) Fig. 1E. Could you label the figure with the number of molecules used in each class average or put this into the legend?
- 4) Supp. Fig. 3: Please increase the font size of the labels and tick labels in this figure such as is seen in Supp. Fig. 2C for example.
- 5) Many of the figures showing cell images could be better labeled. For example, in Fig. 4 it would be useful to have labels at the top of each vertical column to let the reader know which fluorophore is being imaged. This information is not particularly clear in the legends either where on Line 336 it is state "...then individual channels". Particularly in my paper copy the colors are not very distinct.
- 6) Fig. 5E: The legend states that information is given in the Methods. I would prefer that a description of the colors be reiterated in the legend itself to keep the reviewer from having to go back and forth. In reading the appropriate section of the Methods, the colors described do not always seem to match what is seen in the figure itself. Specifically, the light chains are said to be in pink, but in the figure I see several colors for the light chains. Also, it would be good to explain why the model has placed the motor domains at 31 nm distance apart along with the appropriate references.
- 7) This gives rise to a question that might merit discussion. Most of the mechanical stepping data with Myo6 fragments have been generated using a fragment that is dimerized by placing a GCN4 leucine

zipper (presumably parallel coiled-coil) at Arg 994. Do you believe that this construct also forms an antiparallel coiled coil in the sequence proximal to that before becoming antiparallel at the GCN sequence or would you expect that the leucine zipper “templates” a longer stretch of parallel coiled-coil from parts of the Myo6 structure?

Reviewer #2 (Remarks to the Author):

This manuscript investigates the mechanisms of the active (on)/in-active (off) state of myosin VI. Three binding partners, GIPC1, Dab2, & TOM1 are examined the activity of Myo6 as well as the mechanisms of dimerization of myo6 that was poorly understood. Authors used many different approaches, e.g. experimental techniques to answer their questions. Large number of data set were explained sufficiently their model and developed a convincing story. Thus, the manuscript makes a valuable contribution to the research field, especially molecular motor myosin. I have only minor questions.

Minor

1. Line 161-163: authors stated that the Pi release rate of Jo-Myo6-in (54 /s) is lower than that of FLMyo6 WT (63 /s) in Sup. Fig. 2D table. These numbers (63 and 54 /s) of Pi release rates are not significantly difference, e.g. almost the same as a kinetics study and not greatly slow.
2. Figure 1 C and F might be convinced.
3. Please clarify what do authors use the statistic analysis as Standard Deviation (S.D.) or the Standard Deviation of the mean (SEM) in all table and text.
4. For ATPase experiments, please show the concentration of Myo6. In Material and Methods, it says 15- μ M. Does the Myo6 concentration fixed or changed the case of Myo6 protein?
5. ADP.AIF4: For this ATP analogue, I can understand to study the state of ATPase cycle in myosin or molecular motor. However, some general readers may not know, but understanding the analog still widely used. Is it possible to explain in the introduction section for what state of myosin with the ATP analogue?
6. If it's possible or reasonable, please show a transmitted light (white light) image of the MNT-1 cells with the melanosome. To investigate the localization of a fluorescently labeled (GFP or mCherry) proteins with the melanosomes, it would be helpful to shows the melanosome position by a light microscope.

7. In figure 5D, what do short bold red and blue lines represent?
8. Line 391-392: the value 32 and 43 micromolar (μM) at peaks, where does that come from? What does it mean in micromolar.
9. Figure 6 legend 1st sentences: "When auto-inhibited, Myo6 can diffuse across actin-rich..." When a protein diffuses in cells, the localization should be random. Why does Myo6 prefer to stay more actin rich area? Is GFP-FLMyo6 localized at the cell periphery in Supplementary Fig. 8 A (top)?

Reviewer #3 (Remarks to the Author):

In this manuscript, the authors utilized multiple approaches to investigate the off-state auto-inhibition mechanism as well as the partner-binding induced activation and dimerization mechanism of Myo6, a crucial disease-associated unconventional myosin motor in mammals. Firstly, using biochemical and EM-based structural methods, they elucidated that the nucleotide-bound Myo6 motor domain can stabilize the back-folded auto-inhibited conformation of Myo6, and generated a 17 Å resolution structural model of the Myo6 off-state, which reveals a potential interaction of Myo6 CBD with both the motor domain and the lever arm of Myo6. Then, they uncovered the differential binding and activation of Myo6 by distinct Myo6-binding partners. Particularly, GIPC1 rather than TOM1 and Dab2 can directly activate Myo6, and triggers proximal dimerization of Myo6. Subsequently, they biochemically and structurally characterized the proximal dimerization domain of Myo6, and determined the crystal structure of the dimeric Myo6(875-940). Finally, their relevant cellular assays proved that the disruption of the proximal dimerization domain of Myo6 compromises endocytosis in HeLa cells. Overall this manuscript is quite strong, and some findings are novel and valuable. However, in the current version of the manuscript some data and conclusions are not so convincing. In order to support the publication in a high-profile journal such as Nature Communications, additional experiments and analyses are required.

Major comments for the authors:

1. As the resolution of the EM structure of Jo-Myo6-In in current manuscript is very limited, therefore the related structural model of the off-state Myo6 is not so reliable. In order to support the structural model of the off-state Myo6 and confirm the position of the GIPC1-binding N-terminal CBD (CBDN) of Myo6 shown in Figure 3A, the authors should seek to purify the Jo-Myo6-In/GIPC1 complex, and determine its EM structure.
2. The motor domain of Myo6 can directly bind to the CBD of Myo6 (as shown in Table 1), and is directly involved in the stabilization of the Myo6 off-state. Given that the CBD of Myo6 contains two different

modules, the GIPC1-binding CBDN and TOM1/Dab2-binding CBDC, which module of Myo6 CBD is responsible for the interaction with Myo6 motor domain should be further elucidated.

3. The crystallographic section for the crystal structure of the Myo6 dimerization motif needs further improvement. The current statistics in Table S1 are incomplete, and many statistic values (such as Ramachandran plot values and RMSD bonds, angles) as well as some statistic values for the highest resolution shell are missing. Meanwhile, a R value of 30.4% at 2.22 Å resolution for the crystal structure of Myo6(875-940) is surprisingly high. Further refinement is needed.

4. Based on the structure of Myo6 3HB, the L926 residue of Myo6 is not involved in the folding of Myo6 3HB. Furthermore, the L926 residue is also not directly involved in the dimerization of Myo6(875-940) as indicated in the Myo6(875-940) structure (Fig. 5A). Therefore, logically, it is puzzled how the deafness-associated L926Q mutation could impair Myo6 dimerization by perturbing the unfolding of the 3HB. Actually, the SEC-MALS data of Myo6(875-940) shown in Fig. 5D indicates that the Myo6(875-940) dimer is not a stable dimer in solution. Meanwhile, the C-terminal part of 3HB helix-1 and the N-terminal part of 3HB helix-2 contain many hydrophobic residues, once 3HB is unfold, these hydrophobic regions should also be stabilized by some ways. Given the current anti-parallel dimeric structure of Myo6(875-940) (Fig. 5A and 5E), the boundary of the Myo6 proximal dimerization region should be further carefully mapped, especially the N-terminal boundary. The authors should also analyze the dimerization properties of long fragments of Myo6, such as Myo6(861-940) and Myo6(844-940), by SEC-MALS to test whether they can form more stable dimers. The crucial L926 residue may directly participate in the dimerization of Myo6 in these long fragments.

5. As introduced by the authors, NDP52 (or Optineurin) can also specifically bind to the N-terminal CBD (CBDN) of Myo6, in addition to GIPC1. Does the activation of the back-folded form of Myo6 by the GIPC1-binding is a common mechanism shared by other Myo6 CBDN-binding partners? It will be interesting to know whether NDP52 (or Optineurin) could also promote the activation of the back-folded form of Myo6.

6. As shown in Fig 3A, the GIPC1-binding CBDN of Myo6 is far away from the 3HB region, how does GIPC1 bind to the Myo6 CBDN to unfold the 3HB of Myo6? Especially, based on a previous study (eLife. 2017, 6:e27322), GIPC1 has two distinct binding interfaces for Myo6. The authors should provide more direct evidences to support their conclusion that “GIPC1 binds to the Myo6 CBDN to unfold the 3HB of Myo6”, and discuss the related mechanism in the manuscript.

Minor comments for the authors:

(1) The supplementary text file should include page numbers.

(2) Page 2, line 68, “Myo10” should be changed to “Myo7a”.

(3) Page 4, the “CBC” in Figure 1G should be changed to “CBD”; the “CDB” in Figure 1H should be changed to “CDB”; the distance label “17.5 Å and 3.8 Å” in Figure 1H should be defined.

(4) The panel label B and C in the supplementary Figure 4 should be revised, and the corresponding supplementary figure citation in Page 6 of the main text should be also changed.

(5) Page 12, in the left panel of Figure 5A, the N- and C-terminal of Myo6(875-940) should be labeled and indicated. Furthermore, the dimerization interface of Myo6(875-940) in the right panel of Figure 5A should be carefully analyzed and the interface residues should be fully labeled. The side chains of some residues that are not involved in the interactions should be removed.

(6) In the supplementary Figure 5, the protein marker columns should be indicated, and the corresponding arrows for different protein bands should be marked correctly. In addition, for the protein labels, the “147kDa, 33kDa, 17kDa, 12kDa” should be changed to “147 kDa, 33 kDa, 17 kDa, 12 kDa”.

(7) Page 11, line 379, “polar contacts” should be changed to “polar interactions”.

(8) Page 13, line 408, “their ability” should be changed to “their abilities”; line 410, “disturb the bundle” should be changed to “disturb the bundle formation”.

Reviewer #4 (Remarks to the Author):

This paper investigates several structural aspects of myosin six motor protein and how it is regulated by several of its partner proteins. The paper should be revised as explained below.

The paper includes many acronyms that should be defined and explained.

The SEC-SAXS analysis is insufficient.

The authors did not show most of the data and analysis. The units in Supporting figure 1C are incorrect.

The correct way is to show the original scattering data (I vs. q) at several points along the elution curve and then model the original scattering curves. In the current data presentation, it is hard to see what is going on, even if the gyration radius (R_g) is the relevant analysis and the analysis was done for the relevant q -range. The author should present and analyze their data as was done in the following examples:

<https://scripts.iucr.org/cgi-bin/paper?vg5038>

<https://scripts.iucr.org/cgi-bin/paper?jv5008>

<https://onlinelibrary.wiley.com/doi/full/10.1002/pro.4237>

The analysis in Sup Fig 1B is unclear. Please provide more details. How were the values obtained?

The Y-axis legend is incorrect in Figure 1B and Sup Fig 2B. It should be as in the caption of Sup Fig 2B.

Sup Fig 2D: Please show raw data, provide all the details about the stopped-flow experiment, present data analysis, explain it, and only then show the summarizing table.

Figure 1C, H, and Sup Fig 2E: Show the SAXS data analysis and the fit to the entire SAXS curve. Explain how it was done and the expected error of the study; Discuss alternative models that could equally fit the data and show a cluster of such models. Finally, compute the R_g of these models and compare them with the R_g analysis in Sup Fig 1.

Based on the density map, compute the expected SAXS curve and compare it with the SAXS data and the earlier analysis. Then compute and add the expected SAXS curve from each PDB, compare it with the SAXS data, and then discuss the differences.

Why was cryo-TEM not done? A negative stain is less accurate. Please do cryo-TEM single particle analysis.

The K_D analysis is not explained in Sup Fig 3. Many more details are required. The legends are not explained, and the Y-axis units are not defined. The error analysis and the model to which the data fit should be described. The authors should calculate the standard Gibbs free energies associated with their K_D values and add them to Table 1.

Please do it for other K_D values.

Answers to reviewer's Comments:

Reviewer #1 (Remarks to the Author)

The manuscript by Canon et al provides a support for model of myosin 6 compact off state and how the molecule might dimerize via an antiparallel coiled coil that is formed upon opening of the well described three helix bundle motif in this myosin. The SAX model for the back-folded off state is explained by placing various known crystal structures of myosin 6 fragments and speculating on what part of the myosin structure could fill the remaining unaccounted for spaces in the SAX envelop. Several different methodologies are used to generate and support these models including SAXS, protein:protein binding experiments, size exclusion chromatography, enzymatic assays, crystallography and intercellular localization experiments. For the most part the data are well presented and convincing. The results will be of great interest to structural biologists, cell biologists and the communities generally interested in myosin structure and function as well as the deafness.

We thank reviewer 1 for the positive feedback on the manuscript.

I do have some questions and issues that should be addressed.

1) The model shown in in Fig. 6A and in many places in the text assumes that the back-folded conformation of myosin 6 does not bind to actin with high affinity. The covalently stabilized Jo-myosin 6-In construct seems to be an ideal fragment to use in actin sedimentation experiments to quantify this affinity. Could such an experiment be provided?

Following the reviewer's suggestion, we performed co-sedimentations assays where 1 μM of different Myo6 constructs were incubated with 10 μM F-actin in 20 mM Hepes pH 7.5; 1 mM MgCl_2 ; 20 mM KCl; 1 mM ADP; 2 mM NaVO_4 ; 1 mM DTT (Rebuttal Figure 1). Quantification of the bands from SDS-PAGE (below; "In" = Input; "S" = supernatant after centrifugation at 100 000 g; "P" = precipitate; the gel was stained with Readyblue (Sigma) and scanned with a Chemidoc (BioRad). The band corresponding to actin in each lane was used as 100% reference for obtaining the relative myosin quantities by means of the ImageLab software (BioRad). The results show that in the ADP. VO_4 state, Jo-Myo6-In behaves just like FLMyo6: it binds to actin with weaker affinity than Myo6 MD^{Ins2} showing that in the ADP. VO_4 state, the presence of the Tail decreases Myo6 Motor Domain affinity for F-actin. To further characterize the ability of the Jo-Myo6-In construct to interact with F-actin, the new version of the manuscript includes an actin-activated ATPase assay performed with Jo-Myo6-In from which an apparent affinity (K_{ATPase}) of 7 μM of myosin for actin was measured (see Sup Fig 2E). K_{ATPase} was thus in the same range for Jo-Myo6-In and Myo6 (3-18 μM ; De La Cruz *et al.* 2001, doi: 10.1074/jbc.M104136200). Taken together with the negative staining and ATPase experiments shown in the manuscript, this suggests that the Jo-In fusion does not completely lock the movement of the molecule, but impedes the swinging of the lever arm.

Rebuttal Figure 1 – Relative qualitative affinity of different FLMyo6 constructs for F-actin.

The new Sup Fig. 2E (page 4) present the new actin-activated ATPase experiment as follows:

Sup Fig 2

(E) Actin-activated ATPase rate of Jo-Myo6-In as a function of actin concentration. Mean values \pm standard deviation are plotted. ATPase measurements were performed as outlined in **Methods. The V_{max} extrapolates to $\sim 0.08 \text{ sec}^{-1}$ and the K_{ATPase} (apparent actin affinity) is $\sim 7 \mu\text{M}$ actin ($\Delta G^\circ \sim -7 \text{ kcal/mol}$).**

This new actin-activated ATPase is cited in the main text as follows:

Main Text I. 168-170 Actin-activated ATPase measurements revealed a very slow steady state turnover rate for Jo-Myo6-In compared to earlier measurements on wild-type Myo6³⁸, indicating that the conformational changes required to cycle on actin were greatly slowed (Sup Fig. 2E).

2) Line 133: Show a field of negatively-stained FLMyo6 in ADP.VO4 in the supplementary figures for comparison.

The image was added in Sup Fig. 2A (Supplementary information – page 4).

This new supplementary figure is cited in the main text as follows:

Main Text I. 133-135: To further characterize the Myo6 off-state, negative staining electron microscopy (EM) of FLMyo6 in ADP.VO₄ (ADP.Pi analog) (Sup Fig. 2A) resulted in heterogeneous 2D classes, likely due to the intrinsic flexibility of the protein particles.

Sup Fig. 2 – (A) Negative staining image of FLMyo6 without the Jo-In fusion. Purified FLMyo6 was diluted to 50 $\mu\text{g/ml}$ in buffer containing MgADP.VO₄ and used for preparing negative staining grids with 2 % uranyl acetate following standard procedure. Images were collected with a Tecnai G2 at 200 kV.

3) Fig. 1E. Could you label the figure with the number of molecules used in each class average or put this into the legend?

This information was added as follows in the Figure 1 legend:

Main Text I.152-154: (E) Example of a negative staining micrograph of Jo-Myo6-In in ADP.VO₄, with examples of 2D classes (from left to right: 8630; 9284; 9261; 7822 and 7179 particles averaged, respectively).

4) Supp. Fig. 3: Please increase the font size of the labels and tick labels in this figure such as is seen in Supp. Fig. 2C for example.

The font size of the labels and tick labels of Sup. Fig. 3 have been increased as required.

5) Many of the figures showing cell images could be better labeled. For example, in Fig. 4 it would be useful to have labels at the top of each vertical column to let the reader know which fluorophore is being imaged. This information is not particularly clear in the legends either where on Line 336 it is state "...then individual channels". Particularly in my paper copy the colors are not very distinct.

Thank you for this comment. We added a label on each channel of Sup. Fig. 5, 8, 9, 10 (named Sup. Fig. 4, 7, 8, 9 in the initial version of the manuscript) and Fig. 4 to make the figures clearer.

6) Fig. 5E: The legend states that information is given in the Methods. I would prefer that a description of the colors be reiterated in the legend itself to keep the reviewer from having to go back and forth. In reading the appropriate section of the Methods, the colors described do not always seem to match what is seen in the figure itself. Specifically, the light chains are said to be in pink, but in the figure I see several

colors for the light chains. Also, it would be good to explain why the model has placed the motor domains at 31 nm distance apart along with the appropriate references.

We thank the reviewer for this remark. In order to minimize confusion, we added the indications for the light chains on both sides, directly in Fig. 5E. We placed the motor domains 31 nm apart from each other because it is a preferred actin-binding site, based on an average step size for the truncated constructs (at 991 or 1050), the zippered dimer and the full-length protein of ~30 nm (Park *et al.*, 2006; doi: 10.1016/j.molcel.2005.12.015)

This information is now added in the Methods (**Main text I. 757-760**):

Initial text:

The cryoEM structure of Myo6 bound to actin (PDB: 6BNP⁶⁹) was used as basis for placing two Myo6 Motor domains in rigor and PPS states (grey) at the desired 31 nm distance (Fig. 5E).

Modified text:

The cryo-EM structure of Myo6 bound to actin (PDB: 6BNP⁶⁹) was used as basis for placing two Myo6 Motor domains in rigor and PPS states (grey) at a distance compatible with the average step size of ~30 nm previously measured for the truncated constructs (at 991 or 1050), the zippered dimer and the full-length protein^{12,46} (Fig. 5E).

7) This gives rise to a question that might merit discussion. Most of the mechanical stepping data with Myo6 fragments have been generated using a fragment that is dimerized by placing a GCN4 leucine zipper (presumably parallel coiled-coil) at Arg 994. Do you believe that this construct also forms an antiparallel coiled coil in the sequence proximal to that before becoming antiparallel at the GCN sequence or would you expect that the leucine zipper “templates” a longer stretch of parallel coiled-coil from parts of the Myo6 structure?

The initial design of the zippered HMM leucine zipper (Park *et al.*, 2006; doi: 10.1016/j.molcel.2005.12.015) was inserted at sufficient distance from the predicted coiled coil, which in the reality was largely SAH, to not interfere with the phasing. The leucine zipper position did not alter the observed step size of the zippered construct, as compared to the full-length protein (Park *et al.*, 2006; doi: 10.1016/j.molcel.2005.12.015). Later experiments (Mukherjea *et al.*, 2009; doi: 10.1016/j.molcel.2009.07.010) examined the helical content of peptides (using circular dichroism) and their ability to form interactions, such as in bundles and coiled coils, via cooperative melting curves. When we examined a motor-less zippered dimer, we could detect the presence of the GCN4 (melting cooperatively at >90°C) in the zippered HMM, along with the unfolding of the three-helix bundle. We also noted a cooperative melting at ~77°C, which we commented may arise from the presence of a short segment of coiled coil in addition to GCN4. We now conclude that this melting was coming from the anti-parallel coiled coil described in this study, providing further evidence that it is not perturbed by the distal position of GCN4.

Reviewer #2 (Remarks to the Author)

This manuscript investigates the mechanisms of the active (on)/in-active (off) state of myosin VI. Three binding partners, GIPC1, Dab2, & TOM1 are examined the activity of Myo6 as well as the mechanisms of dimerization of myo6 that was poorly understood. Authors used many different approaches, e.g. experimental techniques to answer their questions. Large number of data set were explained sufficiently

their model and developed a convincing story. Thus, the manuscript makes a valuable contribution to the research field, especially molecular motor myosin. I have only minor questions.

We thank Reviewer 2 for the positive remarks on the manuscript.

Minor

1. Line 161-163: authors stated that the Pi release rate of Jo-Myo6-in (54 /s) is lower than that of FLMyo6 WT (63 /s) in Sup. Fig. 2D table. These numbers (63 and 54 /s) of Pi release rates are not significantly difference, e.g. almost the same as a kinetics study and not greatly slow.

The rate of P_i release is indeed similar for the two constructs. What was reduced is the amplitude of the release (*i.e.* amount of phosphate released) (0.15 ± 0.05 for FLMyo6 and 0.05 ± 0.006 for Jo-Myo6-In) as shown in the (initial Sup Fig. 2D), which has now been replaced with ATPase data over a broad range of actin concentrations (new Sup Fig. 2E). The data suggests that there is a very slow rate of ATPase activity, which may result in the reduced amplitude we initially observed in the stopped flow experiments.

The new Sup Fig. 2E (page 4) presents the new actin-activated ATPase experiment as follows:

Sup Fig 2

(E) Actin-activated ATPase rate of Jo-Myo6-In as a function of actin concentration. Mean values \pm standard deviation are plotted. ATPase measurements were performed as outlined in **Methods**. The V_{max} extrapolates to $\sim 0.08 \text{ sec}^{-1}$ and the K_{ATPase} (apparent actin affinity) is $\sim 7 \mu\text{M}$ actin ($\Delta G^\circ \sim -7 \text{ kcal/mol}$).

We have changed the sentence in the main text to introduce this new experiment (**Main text I. 168-170**):

Initial text:

An actin-activated stopped-flow experiment revealed a low P_i release rate for Jo-Myo6-In compared to FLMyo6 wild-type (WT), indicating that the conformational changes required to release P_i were greatly slowed (Sup Fig. 2D).

Modified text:

Actin-activated ATPase measurements revealed a very slow steady state turnover rate for Jo-Myo6-In compared to earlier measurements on wild-type Myo6³⁸, indicating that the conformational changes required to cycle on actin were greatly slowed (Sup Fig. 2E).

2. Figure 1 C and F might be convinced.

To make a clearer comparison of our SAXS envelope and our negative staining 3D-reconstruction, a superimposition of the two has been added in Sup Data 2 (Supplementary informations – page 28):

Sup Data 2 - FLMyo6 scattering data interpretation

(C) SAXS envelope of FLMyo6 (grey) was manually docked into the negative staining 3D-reconstruction of the Jo-Myo6-In (deep blue). This comparison shows similar size and shape for both although the negative staining 3D-reconstruction displays more details and an additional blob corresponding to the Jo/In domain.

The Supplementary Data is cited in the main text as follows:

Main text I.156-157: Negative staining 3D reconstruction and the ab initio SAXS envelope exhibit similar overall size and shape for Myo6 (Sup Data 2C).

Main text I.173: The 3D reconstruction of the Myo6 off-state at ~ 17 Å resolution (Fig. 1F, Sup Movie 1) is consistent in shape and dimensions with SAXS data of FLMyo6 (Fig. 1C, F; Sup Data 2C).

Main text I. 693-696: The complete model of the Myo6 off-state is presented in Fig. 3A. Placement of the different domains is further supported by the crosslink experiment presented in Sup Fig. 4 and Sup Table 1. Its ability to fit our off-state Myo6 SAXS data was tested (Sup Data 2D-F and Sup Data 3B-C).

We realize now that the Figure 1 we had included in the first version can lead to confusion when both the SAXS envelope (Fig. 1C) and the EM density map (Fig. 1H) are presented side by side. SAXS was used to identify the overall shape of the off-state and the best conditions to stabilize it. Together with our microscale thermophoresis experiments (Table 1), the SAXS envelop allowed us to design the Jo-Myo6-In construct, which is also coherent with FRET experiments published by the Toseland lab when the N and C-termini of FLMyo6 were shown to be in proximity when the motor adopts an off-state. The negative staining 3D reconstruction of Jo-Myo6-In then

provided us the means to interpret the map with PDB coordinates, taking into account all biochemical information we also had gathered, as described in the main text. The SAXS data was not used together with the EM data to dock the coordinates. As suggested by Reviewer 4, we can further validate the EM model by the fact that it is consistent with SAXS measurements that we have gathered.

To avoid confusion between SAXS and EM data, the SAXS envelope is now presented in green, and the **Figure 1** legend was modified as follows:

Main text I.149-151: (C) Representation of the *ab initio* SAXS envelope of Myo6 in ADP.AIF₄ condition (green) with MD^{Ins2-IQ-3HB} docked. Myo6 adopts a compact conformation that requires Myo6 to fold back after the 3HB domain (See Methods and Sup Data 2A-B).

Main text I. 154-157: (F) EM density for Jo-Myo6-In (grey mesh) obtained by negative staining. Coordinates of Myo6 fragments and Jo-In were manually docked inside the negative staining 3D reconstruction (see Methods). Negative staining 3D-reconstruction and the *ab initio* SAXS envelop exhibit similar overall size and shape for Myo6 (Sup Data 2C).

Main text I. 161-162: (H) CBD^c (brown) added to the negative staining based model pictured in (F), (see Methods).

And the Methods were modified as follows:

Main text I. 674-675: We first positioned the Motor domain-Lever arm (residues 1-917) in the Jo-Myo6-In maps from SAXS and negative staining EM.

3. Please clarify what do authors use the statistic analysis as Standard Deviation (S.D.) or the Standard Deviation of the mean (SEM) in all table and text.

We added the following text (highlighted in yellow) to clarify this point:

Main text:

Main text I.119: Radius of gyration (R_g) = 49.23 ± 0.92 (standard deviation; SD) Å

Main text I.122: $R_g = 84.18 \pm 4.33$ (SD) Å

Main text I.204: Dissociation constant (K_D) $\pm K_D$ confidence (with a 68% confidence using the NTAanalysis software)

Main text I.226-227: Both FLMyo6 (L926Q) and FLMyo6 (SAHmimic) mutants display higher ATPase rates (2.86 ± 0.12 (SD) and 4.83 ± 0.11 (SD) $s^{-1} \cdot \text{Head}^{-1}$ respectively) than the wild-type (0.65 ± 0.08 (SD) $s^{-1} \cdot \text{Head}^{-1}$)

Main text I.251: Error bars = SD

Main text I.252: Error bars = SD

Main text I.296: Error bars = SD

Supplementary information:

Supplementary I.49-50: R_g value \pm standard deviation is depicted for each condition.

Supplementary I.52: Error bars = standard deviation.

Supplementary I.84-85: Mean values \pm standard deviation are plotted.

Supplementary I.95-96: Error bars = standard deviation.

Supplementary I.140: Error bars = SEM.

Supplementary I.149: *Dissociation constant (K_D) \pm K_D confidence (with a 68% confidence using the NAnalysis software)*

Supplementary I.181: *Error bars = SEM.*

4. For ATPase experiments, please show the concentration of Myo6. In Material and Methods, it says 15- μ M. Does the Myo6 concentration fixed or changed the case of Myo6 protein?

The concentration of Myo6 is fixed at 150 nM in all our ATPase experiments. To clarify this point, the main text was rephrased (**Main text I.646-647**):

Initial text:

F-Actin was used at 40 μ M and Myo6 at 150 nM with 2.5 μ M additional CaM in all our experiments.

Modified text:

Myo6 was used at 150 nM, F-Actin was used at 40 μ M (unless otherwise noted) and 2.5 μ M additional CaM were added in all our experiments.

5. ADP.AIF4: For this ATP analogue, I can understand to study the state of ATPase cycle in myosin or molecular motor. However, some general readers may not know, but understanding the analog still widely used. Is it possible to explain in the introduction section for what state of myosin with the ATP analogue?

We added the following text in legend to clarify this point:

Main text I.145-146: *ADP.AIF₄ (a widely used ADP.P_i analog that stabilizes the pre-powerstroke of Myo6)*

Main text I.247 and I.250: *ADP.AIF₄ (ADP.P_i analog)*

6. If it's possible or reasonable, please show a transmitted light (white light) image of the MNT-1 cells with the melanosome. To investigate the localization of a fluorescently labeled (GFP or mCherry) proteins with the melanosomes, it would be helpful to shows the melanosome position by a light microscope.

We provide here the images required by the reviewer:

Rebuttal Figure 2 – Melanosomes detection in MNT-1 cells

(Left) BrightField image of representative MNT-1 cells co-expressing GFP^{FLMyo6} (wild-type) with $mcherry^{MST}$ and $iRFP^{VAMP7}$. A “Z projection” (min intensity) has been applied. **(Right)** Melanin pigments were detected as black spots by the FIJI macro. For each ones, a circle diameter of $0.350\mu m$ centered on the detected pigment is generated. Threshold: 3000. Diameter: XY $0.075\mu m$ Z: $0.2\mu m$

7. In figure 5D, what do short bold red and blue lines represent?
and

8. Line 391-392: the value 32 and 43 micromolar (μM) at peaks, where does that come from? What does it mean in micromolar.

To answer both comments 7 and 8, we completed the legend of **Fig. 5** with information to help the readers who are not familiar with SEC-MALS profiles, as follows (**Main text 1.392-399**):

Initial text:

(D) SEC-MALS profiles of Myo6 875-940 WT (red) and T888D.R892E.V903D mutant (blue), following injection of $50\mu l$ at 10 mg/mL in $10\text{ mM Tris-HCl pH }7.5$; 50 mM NaCl ; 5 mM NaN_3 ; 0.5 mM TCEP . WT elutes as dimers ($32\mu M$ at the peak) and T888D.R892E.V903D mutant elutes as monomers ($43\mu M$ at the peak). Measured molecular weight is heterogeneous due to the small size of the peptides through elution (low light scattering signal), but at the peaks it coincides with the expected masses

Modified text:

(D) SEC-MALS profiles of Myo6 875-940 WT (red) and T888D.R892E.V903D mutant (blue), following injection of $50\mu l$ at 10 mg/mL in $10\text{ mM Tris-HCl pH }7.5$; 50 mM NaCl ; 5 mM NaN_3 ; 0.5 mM TCEP . Thin lines: static light scattering; thick lines: measured molecular mass. WT elutes as dimers ($32\mu M$ concentration at the peak, as measured by the in-line refractometer) and T888D.R892E.V903D mutant elutes as monomers ($43\mu M$ at the peak). Due to the small size of the peptides, the measured molecular weight is heterogeneous through elution (low light scattering signal) but coincides with the expected masses at the peaks.

9. Figure 6 legend 1st sentences: “When auto-inhibited, Myo6 can diffuse across actin-rich...” When a protein diffuses in cells, the localization should be random. Why does Myo6 prefer to stay more actin rich area? Is GFP-FLMyo6 localized at the cell periphery in Supplementary Fig. 8 A (top)?

As shown when answering Reviewer 1 question 1, FLMyo6 with an ADP.P_i analog trapped does not bind strongly to F-actin. The proximity of the CBD prevents fast activation of the motor in a constitutive manner and it prevents the ability of the motor to dimerize. However, even when auto-inhibited, Myo6 can undergo facilitated diffusion and concentration in actin-rich regions due to weak transient interactions with F-actin.

We have changed the text of **Fig. 6** in order to clarify this point. This legend now reads as follows: **Main text I. 532-544** (A) *When auto-inhibited, Myo6 can diffuse across actin-rich regions and interacts weakly with F-actin. These weak actin interactions (~7 μM apparent affinity, estimated in Sup Fig 2D) result in facilitated diffusion and in increasing the Myo6 concentration in actin-rich regions of the cell. Once recruited by a partner, Myo6 is activated and starts performing its cellular function.* (B) *Scheme representing possible activation mechanisms for Myo6. Myo6 domains are color coded: Myo6 MD (grey), Ins2/CaM (purple), IQ/CaM (red/pink), 3HB in blue, SAH (green), DT (orange), CBD (brown), and the partner binding sites (garnet). The binding site (WWY) for Dab2 and TOM1 is blocked, preventing recruitment of Myo6 without a prior unfolding signal prior to unblock their binding. GIPC1 can bind the accessible RRL motif resulting in Myo6 recruitment and opening. Other signals can act as unfolding factors such as Ca²⁺, which can allow TOM1 to bind to Myo6. Such an activation cascade was previously proposed³⁶. Once unfolded, Myo6 potentially acts as a monomer, as previously proposed³⁵ upon TOM1 binding; or it can dimerize²⁹ through proximal dimerization, as demonstrated in this study with GIPC1 binding; or it dimerizes through distal dimerization upon Dab2 binding¹⁴, which may lead to proximal dimerization.*

Reviewer #3 (Remarks to the Author)

In this manuscript, the authors utilized multiple approaches to investigate the off-state auto-inhibition mechanism as well as the partner-binding induced activation and dimerization mechanism of Myo6, a crucial disease-associated unconventional myosin motor in mammals. Firstly, using biochemical and EM-based structural methods, they elucidated that the nucleotide-bound Myo6 motor domain can stabilize the back-folded auto-inhibited conformation of Myo6, and generated a 17 Å resolution structural model of the Myo6 off-state, which reveals a potential interaction of Myo6 CBD with both the motor domain and the lever arm of Myo6. Then, they uncovered the differential binding and activation of Myo6 by distinct Myo6-binding partners. Particularly, GIPC1 rather than TOM1 and Dab2 can directly activate Myo6, and triggers proximal dimerization of Myo6. Subsequently, they biochemically and structurally characterized the proximal dimerization domain of Myo6, and determined the crystal structure of the dimeric Myo6 (875-940). Finally, their relevant cellular assays proved that the disruption of the proximal dimerization domain of Myo6 compromises endocytosis in HeLa cells. Overall this manuscript is quite strong, and some findings are novel and valuable.

We appreciate the reviewer’s positive statements as well as the thoughtful suggestions described below.

However, in the current version of the manuscript some data and conclusions are not so convincing. In order to support the publication in a high-profile journal such as Nature Communications, additional experiments and analyses are required.

Major comments for the authors:

1. As the resolution of the EM structure of Jo-Myo6-In in current manuscript is very limited, therefore the related structural model of the off-state Myo6 is not so reliable. In order to support the structural model of the off-state Myo6 and confirm the position of the GIPC1-binding N-terminal CBD (CBDⁿ) of Myo6 shown in Figure 3A, the authors should seek to purify the Jo-Myo6-In/GIPC1 complex, and determine its EM structure.

We thank this reviewer for this suggestion.

In order to consolidate our structural model of the Myo6 off-state, and in particular to confirm the position of the GIPC1-binding N-terminal CBD (CBDⁿ) of Myo6, we carried out a crosslinking mass spectrometry analysis of the FLMyo6 using the disuccinimidyl sulfoxide (DSSO) crosslinker. 213 out of the 247 crosslinks detected were consistent with our structural model (<50 Å distance between Cα). These results are now described in Sup Text (page 21), Sup Fig. 4 (page 7), Sup Table 1 (page 22-23).

The supplementary Text page 21 describes these new results:

To further validate our structural model, we carried out a crosslinking mass spectrometry (XL-MS) analysis of the purified FLMyo6 (Sup Fig. 4A-C, Sup Table 1). 213 out of 247 detected crosslinks are in agreement with the structural model since they correspond to distance between Cα < 50 Å, with 171 crosslinks having a distance less than 30 Å. The SAH can be detected in close proximity of the 3HB and the CaM light chains, further confirming the folding back of the molecule (Fig. 3A, Sup Fig. 4C). Interestingly, a relatively high number of crosslinks were detected between the helices in the 3-helix bundle (Sup Fig. 4B), confirming that the bundle remains closed in the off-state.

Overall, only a few crosslinks were detected within the CBD region and the rest of the molecule. One crosslink was found between the CBD^c and the Motor and one crosslink was identified between the CBD^c and the SAH. Taken together, these contacts situate the CBD^c between the Motor, CaM and SAH as expected in our model.

Finally, two crosslinks were detected between the CBDⁿ and the rest of the Myo6. Both crosslinks are in agreement with our model. Note that overall, only two crosslinks were detected for the CBDⁿ region, even when considering detected distances >50 Å. These results suggest a high flexibility in this region. Interestingly, this flexibility could allow for a higher accessibility of the RRL motif, further confirming our model shown in Fig. 3A.

Together, our data suggests a model in which the tail of Myo6 folds back towards the 3HB, with the SAH in close proximity to the neck region, and the CBD^c nested in a groove between the Lever arm and the Motor only accessible when the molecule is in the pre-powerstroke state.

Additionally, the following sentences were added to the main text to introduce this experiment:

Main text I.198-200: Finally, we performed a crosslinking mass spectrometry analysis of the purified FLMyo6 to validate our structural model (Sup Text; Sup. Fig. 4A-C; Sup. Table 1).

Main text I.282-284: Note that the position of the CBDⁿ is consistent with the crosslinks found between CBDⁿ and the rest of the Myo6 molecule through crosslinking mass spectrometry of the purified FLMyo6 with disuccinimidyl sulfoxide (DSSO) (Sup Text; Sup. Fig. 4A-C; Sup. Table 1).

Main text I.693-695: The complete model of the Myo6 off-state is presented in Fig. 3A. Placement of the different domains is further supported by the crosslink experiment presented in Sup Fig. 4 and Sup Table 1.

The experimental conditions are also described in supplementary methods pages 33-34.

The characterization of a complex between Jo-Myo6-In and GIPC1 by electron microscopy is challenging. The interaction between the Myo6 off-state and GIPC1 must be transient, as our ATPase assays demonstrated that GIPC1's addition to FLMyo6 results in Myo6 opening (Fig. 3E). Thus, obtaining a Jo-Myo6-In/GIPC1 complex stable enough to allow its structure determination at the concentrations required would be very difficult. A large excess of GIPC1 would then be required to ensure Jo-Myo6-In/GIPC1 complex formation, bringing additional difficulties as this would probably produce too much background on the grid to allow structure determination.

Cross-linking of the complex would likely be necessary to ensure complex stability, and this usually requires several rounds of optimization. Such work, in combination with more detailed biophysical and biochemical assays to properly describe the activation by GIPC1 in details would be worth publishing in the future, yet they are outside the scope of the current manuscript.

2. The motor domain of Myo6 can directly bind to the CBD of Myo6 (as shown in Table 1), and is directly involved in the stabilization of the Myo6 off-state. Given that the CBD of Myo6 contains two different modules, the GIPC1-binding CBDⁿ and TOM1/Dab2-binding CBD^c, which module of Myo6 CBD is responsible for the interaction with Myo6 motor domain should be further elucidated.

We thank reviewer 3 for this proposition, as indeed it would be interesting to know the relative contribution of CBD^c and CBDⁿ. It appears that the most direct way would be to measure the interactions of the Motor domain with distinct fragments of the CBD, (CBDⁿ and CBD^c separately). However, our preliminary experiments are indicating that such experiments might not provide a definitive answer for understanding the precise role of the CBDⁿ relative to the CBD^c in maintaining the Myo6 off-state:

Our exploration of the GIPC1 binding site in the context of the whole CBD provides different results than those previously reported by Shang *et al.*, 2017; doi: 10.7554/eLife.27322, when they measured the interaction of GIPC1 with a Myo6 Tail fragment limited to the CBDⁿ (See text below in answering the comment 6 of this reviewer). Moreover, we have evidence that the CBDⁿ domain behaves poorly when expressed alone (See text below concerning our preliminary evidence regarding the CBDⁿ) as it tends to oligomerize and precipitate.

For those reasons, we doubt that when expressed alone, the explored CBDⁿ structural states are representative of those that this part of the Myo6 Tail explores in the context of the full Tail. In short,

the CBDⁿ contribution must be explored in the context of the whole CBD or even longer fragments of the Myo6 molecule, as done in our manuscript.

Table 1 demonstrates the key role of the CBD^c in the stabilization of the Myo6 off-state, since 4 point mutations within a CBD^c loop are sufficient to disrupt the interaction of the whole CBD with the MD^{Ins2/IQ/3HB} but those data do not exclude the participation of the CBDⁿ in the stabilization of the off-state. Thereby, the crosslinking mass spectrometry experiment (introduced in the response of question 1 of Reviewer 3) suggests a rather close (< 50 Å) proximity between the CBDⁿ and the rest of the Myo6 molecule (contacts detected with the Motor domain and the SAH) (**Sup Text; Sup Fig. 4; Sup Table 1**). Note that a previous study has proposed that the ability of CBDⁿ to interact with CaM in a calcium concentration-dependent manner would be key for stabilizing the off-state (**Batters *et al.*, 2016; doi: 10.1073/pnas.1519435113**). In their study, these authors did not detect a role for the CBD^c to bind the MD^{Ins2/IQ/3HB} construct. However, they used a F-actin pull down assay and thus imposed a rigor structural state for the Myo6 Head (nucleotide free state) while testing the contribution of the CBD^c. Thus, their assay is in agreement with our findings, but concluded wrongly that the CBD^c was not important for the interaction as we have shown in **Fig.1B, Table 1 and Sup Fig. 1** that the MD^{Ins2/IQ/3HB} required nucleotide (*i.e.* to adopt a PPS state) for the interaction with the CBD^c to occur. Concerning the experiments done to assess how CaM would bind the CBDⁿ, **Batters *et al.*, 2016** used in fact a complete CBD and concluded that only CBDⁿ would be involved in interaction due to the previous conclusion that CBD^c was not involved in binding of the head. Our XL-MS data (**Sup Table 1**) provides only few crosslinks of CBDⁿ or CBD^c with the rest of the molecule. The crosslinking distances reported for these domains (**Sup Table 1**) are consistent with the model shown **Fig. 3A**.

A more precise answer to this reviewer's comment must await a higher-resolution structure of FLMyo6, which has not been possible in the context of this study, as the FLMyo6 flexibility makes this a difficult structure to solve even after addition of the Jo and In tags. We thus think that answering fully this question needs to await additional structural data, but this is beyond the scope of the current study.

Preliminary evidence regarding the CBDⁿ

The currently available structural information for the CBDⁿ indicates that it is likely heavily influenced by its surrounding. In particular, **Shang *et al.*, 2017; doi: 10.7554/eLife.27322**, showed that CBDⁿ tends to oligomerize when it is produced alone. This is also what we have observed no matter the limits of the CBDⁿ and the buffer conditions explored. In addition, the CBDⁿ structure is only partially resolved and the available structures highlight the high flexibility of this domain when produced alone (**He *et al.*, 2016; doi: 10.1016/j.celrep.2016.01.079**). It is thus likely that measuring the ability of the CBDⁿ to interact with the rest of the protein would provide only partial information, possibly heavily biased information as we cannot exclude that CBD^c or other surrounding regions would be required for the CBDⁿ to select the structural state that is adopted while it is involved in intra-molecular interactions.

3. The crystallographic section for the crystal structure of the Myo6 dimerization motif needs further improvement. The current statistics in Table S1 are incomplete, and many statistic values (such as Ramachandran plot values and RMSD bonds, angles) as well as some statistic values for the highest resolution shell are missing. Meanwhile, a R value of 30.4% at 2.22 Å resolution for the crystal structure of Myo6(875-940) is surprisingly high. Further refinement is needed.

We thank the reviewer for pointing out the difficulties in obtaining reliable crystallographic structures of short coiled-coil fragments. Those difficulties explain, at least partially, the rarity of purely coiled-coil protein structures in the Protein Data Bank. In the case of our structure of the Myo6 dimerization

motif 8ARD, it was obtained after (literally) hundreds of crystals that had been tested with different fragments of Myo6 and different crystallization, freezing and annealing conditions over several months of work. Several rounds of refinement and construction were necessary for convergence, knowing that small differences in positioning of atoms in a 62 residues peptide leads to considerable differences in R/Rfree statistics. Upon deposition in the PDB, the unusual statistics listed by reviewer #3 were outlined as values to be verified, and not as major issues that would impede the structure from becoming publicly available. We use the 2Fo-Fc map produced by Buster as shown in Fig. 5B and Sup Fig. 13, together with the geometry assessment to judge the quality of our structure. We chose to maintain flexible side chains for which there was no density to guide construction – simply because we have no reason to believe that the corresponding atoms are not present in the crystal – and, given the inherent nature of the coiled-coil fragment and its crystal packing, this implies 21% of the residues. By consequence, the impact on Rfree is important. Canonical 20-60 kDa, globular proteins that comprise the majority of protein structures found in the PDB crystallize in more favorable lattices, giving lower Rwork and Rfree final values, from which the “rule of thumb” of 10 % resolution (*i. e.* Rfree *needs* to be 0.2 for structures at 2 angstroms resolution; 0.3 for 3 angstroms and so on) was derived. However, the intended usefulness of Rwork and Rfree is limited to the very own dataset different models are being confronted to, and not to be compared between different structures (see Wang, 2015 - doi: 10.1002/pro.2639 – as an example of relevant discussion on the topic).

However, we agree that the previous Sup Table 1 could provide additional information. We thus now completed this table as requested, and added anisotropy-related statistics that we believe useful. We thus replaced the previous table with the following Sup Table 2:

Diffraction limits (Å) and corresponding principal axes of the ellipsoid fitted to the diffraction cut-off surface as direction cosines in the orthogonal basis (standard PDB convention), and in terms of reciprocal unit-cell vectors:

Diffraction limit #1:	2.669	(1.0000, 0.0000, 0.0000)	0.894 a* - 0.447 b*
Diffraction limit #2:	2.669	(0.0000, 1.0000, 0.0000)	b*
Diffraction limit #3:	2.079	(0.0000, 0.0000, 1.0000)	c*

Eigenvalues of overall anisotropy tensor on $|F|s$ (Å²) and corresponding eigenvectors of the overall anisotropy tensor as direction cosines in the orthogonal basis (standard PDB convention), and in terms of reciprocal unit-cell vectors:

Eigenvalue #1:	81.56	(1.0000, 0.0000, 0.0000)	0.894 a* - 0.447 b*
Eigenvalue #2:	81.56	(0.0000, 1.0000, 0.0000)	b*
Eigenvalue #3:	28.46	(0.0000, 0.0000, 1.0000)	c*

Statistics for the highest-resolution shell are shown in parentheses:

Wavelength (Å)	0.97857
Resolution range (Å)	25.45 - 2.219 (2.298 - 2.219)
Space group	P 6 ₅ 2 2
Unit cell	29.387 Å 29.387 Å 295.649 Å 90° 90°
Total reflections	8883 (825)

Unique reflections	4442 (46)
Multiplicity	2.0 (2.0)
Completeness (%)	66.67 (11.27)
Mean I/sigma(I)	9.38 (0.10)
Wilson B-factor (Å²)	35.75
R_{merge}	0.01821 (2.691)
R_{meas}	0.02576 (3.805)
R_{pim}	0.01821 (2.691)
CC_{1/2}	1 (0.495)
CC	1 (0.814)
Reflections used in refinement	2965 (47)
Reflections used for R_{free}	298 (6)
R_{work}	0.3054 (0.4245)
R_{free}	0.3485 (0.3730)
CC_(work)	0.922 (0.512)
CC_(free)	0.856 (0.796)
Number of non-hydrogen atoms	517
macromolecules	512
ligands	0
solvent	5
Protein residues	62
RMS(bonds)	0.012
RMS(angles)	1.43
Ramachandran favored (%)	100.00
Ramachandran allowed (%)	0.00
Ramachandran outliers (%)	0.00
Rotamer outliers (%)	14.04
Clash score	0.96
Average B-factor (Å²)	52.96
macromolecules	53.11
solvent	37.63

4. Based on the structure of Myo6 3HB, the L926 residue of Myo6 is not involved in the folding of Myo6 3HB. Furthermore, the L926 residue is also not directly involved in the dimerization of Myo6(875-940) as indicated in the Myo6(875-940) structure (Fig. 5A). Therefore, logically, it is puzzled how the deafness-associated L926Q mutation could impair Myo6 dimerization by perturbing the unfolding of the 3HB.

As shown in Fig. 5A, L926 is found outside of the proximal dimerization surface once the 3HB is unfolded and the proximal dimeric coiled-coil is formed. Thus, it is unlikely to be involved in direct interactions within this region. However, our ATPase data (Fig. 5F) and our fluorescence quenching data (Table 2) clearly demonstrate that this mutation affects Myo6 dimerization, as we found that introducing the mutant side chain helps keeping the 3HB folded, preventing its ability to participate

in dimerization. In addition, the actin-activated ATPase data of FLMyo6 L926Q is more consistent with some of the heads being monomeric, rather than dimeric Fig. 5F. We can only propose suggestions as to how introducing the Glutamine side chain rather than a Leucine at position 926 matters for 3HB dynamics, and for proximal dimerization.

One possibility is that the L926Q mutation alters the dynamics and/or stability of the SAH helix. That could lead to difficulties in reorganizing the different elements involved in forming the dimeric coiled-coil. The 3HB would hence tend to remain folded. As a consequence, equilibrium would tend towards the folding of the 3HB, which would prevent dimerization. The mutation could promote the open metamorphic form rather than the closed form of the 3HB, without promoting the dimerization if the stability of the helices required for dimerization would be changed by the mutation: the FLMyo6 sequestered state would be unfavoured without being able to promote dimerization.

A highly speculative possibility is that the L926 of one Myo6 monomer would interact with another Myo6 monomer once the two have been unfolded and placed in close proximity by a cargo. In this view, L926 would directly destabilize the 3HB of the second Myo6 molecule, rather than the 3HB within the same molecule. This destabilization could occur due to steric hindrance of the 3HB, or due to transient interactions with residues of the neighboring 3HB, due to the clustering of Myo6 molecules induced by cargo binding, or by a membrane. As stated in our previous study, even in the absence of the Motor domain, the 3HB can unfold: circular dichroism (CD) experiments shown in (Mukherjea *et al.*, 2009; doi: 10.1016/j.molcel.2009.07.010), demonstrated that if two headless Myo6 constructs are held together distal to the SAH, then the 3HB is induced to unfold.

Actually, the SEC-MALS data of Myo6(875-940) shown in Fig. 5D indicates that the Myo6(875-940) dimer is not a stable dimer in solution. Meanwhile, the C-terminal part of 3HB helix-1 and the N-terminal part of 3HB helix-2 contain many hydrophobic residues, once 3HB is unfold, these hydrophobic regions should also be stablized by some ways. Given the current anti-parallel dimeric structure of Myo6(875-940) (Fig. 5A and 5E), the boundary of the Myo6 proximal dimerization region should be further carefully mapped, especially the N-terminal boundary. The authors should also analyze the dimerization properties of long fragments of Myo6, such as Myo6(861-940) and Myo6(844-940), by SEC-MALS to test whether they can form more stable dimers.

We agree with the reviewer that the characterization of the region N-terminus of the 3HB is of interest. We designed several constructs to test whether they can participate in proximal dimerization.

To answer about the potential involvement of the N-terminal region in dimerization, MALS results were obtained on Myo6(849-955). This has confirmed the ability of this peptide to form dimers as shown in Rebuttal Figure 3. The affinity of the dimer does not improve compared to 875-955, indicating that the proximal region does not strongly participate in dimerization and that the region that we crystallized contains most of the important interactions to promote proximal dimerization.

Rebuttal Figure 3 – SEC-MALS profiles for His-Myo6(849-955) (blue) and His-Myo6(875-940) (red) are shown. Thin lines: static light scattering signal; thick lines: molecular mass as determined by the software Astra (Wyatt Technologies). 20 μ l of His-Myo6(849-955) at 1.1 mM eluted with 26.15 ± 0.08 kDa (93.4% of the expected mass of the dimer) at 6.4μ M at the peak; whereas 20 μ l of His-Myo6(875-940) at 1.7 mM eluted with 16.3 ± 0.02 kDa (84.9% of the expected mass of the dimer) at 5.99μ M at the peak.

We are aware that the affinities estimated from a SEC profile will certainly be underestimated, since the SEC column is pushing the equilibrium towards dissociation as it separates monomers from dimers during the elution. It is also possible that the presence of upstream and downstream domains further stabilizes the helices – hence dimerization – in the context of active FLMyo6. As Myo6 dimerization is linked to the regulation of on- and off-states, we expect those dimers to form without high affinity.

5. As introduced by the authors, NDP52 (or Optineurin) can also specifically bind to the N-terminal CBD (CBDN) of Myo6, in addition to GIPC1. Does the activation of the back-folded form of Myo6 by the GIPC1-binding is a common mechanism shared by other Myo6 CBDN-binding partners? It will be interesting to know whether NDP52 (or Optineurin) could also promote the activation of the back-folded form of Myo6.

We detect very little activation of FLMyo6 using NDP52, as shown in **Rebuttal Figure 4**.

Rebuttal Figure 4 – Actin activated ATPase activity of FLMyo6 depending on increasing concentrations of NDP52₂₀₂₋₄₄₆ (indicated in μ M). Myo6 was used at 150 nM, F-Actin was used at 40 μ M and 2.5 μ M additional CaM was added.

We can explain this result by the fact that the precise NDP52 binding site is unknown. Indeed, even if GIPC1 and NDP52 both bind the RRL site, in mammalian two-hybrid assays, mutations of RRL into ARL or RRA impact differently Myo6 binding to NDP52 and GIPC1 (Arden *et al.*, 2016; doi: 10.1042/BCJ20160571), which may explain why they have different impact on activation.

Rebuttal Figure 5 – Mammalian two-hybrid assays, extracted from Arden *et al.*, 2016; doi: 10.1042/BCJ20160571. RRA (i.e. L1109A) mutant strongly impairs GIPC1 ability to interact with Myo6 yet the mutation has no significant impact on NDP52 ability to bind Myo6.

Our goal in this manuscript was to show that not all partners are equally recruited and indicate the role of the GIPC1 partner to promote proximal dimerization. We believe that the paper is already quite novel in these concepts and that it is outside the scope of this paper to generalize these concepts with different Myo6 partners, in particular since their binding sites have not been characterized precisely. To avoid any confusion, we removed NDP52 from the introduction of the paper:

Main text I. 78-80: For these cellular functions, Myo6 must associate with different binding partners, such as Dab2, GIPC1 and TOM1 in distinct endosomal compartments²²⁻²⁴ and NDP52 in autophagy and in RNA Polymerase II transcription²⁵.

Main text I. 85-87: TOM1 and Dab2 bind Myo6 through its WWY motif on the C-terminus part of its cargo-binding domain (CBDⁿ; Fig. 1A) while GIPC1 and NDP52 binds the RRL motif on the CBDⁿ (Fig. 1A)

6. As shown in Fig 3A, the GIPC1-binding CBDN of Myo6 is far away from the 3HB region, how does GIPC1 bind to the Myo6 CBDN to unfold the 3HB of Myo6?

One important result in our manuscript describes that GIPC1 promotes proximal dimerization, which implies unfolding of the 3HB. It is worth noting that FRET results using FLMyo6 described in (Rai *et al.*, 2022; doi: 10.1016/j.jbc.2022.101688) also indicate that the 3HB would unfold upon GIPC1 binding. We agree with the reviewer that understanding how GIPC1 binding promotes dimerization is an interesting subject of investigations.

It should be noted that in the off-state conformation, we propose that the GIPC1 binding site (CBDⁿ) is in close proximity to the Lever arm. Thus, we can speculate that GIPC1 binding on CBDⁿ results in a loss of the interactions of the Lever arm with the CBDⁿ, resulting in promoting 3HB unfolding.

However, the system is complex since part of the Myo6 structure is still not determined in between the Distal tail and the CBD, and is likely to change conformation when the active and inactive forms of Myo6 are considered. Thus, answering this question will require many additional experiments, most likely the subject of a whole paper in itself. One important key would be to determine the atomic structure of FLMyo6. Although we have performed cryo-EM and crystallography experiments to do so, they have been difficult, most likely due to the internal flexibility of the Myo6 molecule, preventing access to high resolution and model building. Additionally, studying metamorphic structures is difficult. It is thus the case in particular for regions that include the 3HB up to the Tail that can be monomeric or dimeric depending on the context of the molecule. The structure of this region depends on the presence of other regions of the protein, which are difficult to study by conventional structural approaches. We have been trying without success so far on the description of higher resolution

structures of the Myo6/GIPC1 complex that would allow proximal dimerization to be observed, or fragments of Myo6 that would also describe the Distal domain immediately preceding the CBD.

Especially, based on a previous study (eLife. 2017, 6:e27322), GIPC1 has two distinct binding interfaces for Myo6. The authors should provide more direct evidences to support their conclusion that “GIPC1 binds to the Myo6 CBDN to unfold the 3HB of Myo6”, and discuss the related mechanism in the manuscript.

To describe more completely how GIPC1 binds the CBD, we performed microscale thermophoresis (MST) experiments with the full CBD (Rebuttal Table 1).

In contrast, the results previously described in the paper mentioned by the reviewer (Shang et al 2017; doi: 10.7554/eLife.27322) were obtained using the CBDⁿ, a poorly behaving fragment of the Tail when expressed alone.

The data we have obtained with MST experiments do not confirm the previous results. This is not surprising in light of the remarks exposed in the paragraph “Preliminary evidence regarding the CBDⁿ” already provided to this Reviewer 3 in answer to his request #2.

The GIPC1 binding results we obtained with the larger CBD are consistent in fact with only one of the two binding sites reported in the context of the smaller flexible CBDⁿ construct (Shang et al 2017; doi: 10.7554/eLife.27322). The presence of the CBD^c immediately next to the CBDⁿ improves the quality of the protein expressed, but can also mask part of the CBDⁿ surface, possibly via interactions between CBD^c and CBDⁿ. This suggests that the CBDⁿ problems reported earlier in the paragraph “Preliminary evidence regarding the CBDⁿ” (see response #2 of Reviewer 2) are linked to the fact that CBDⁿ is not a standalone structural subdomain and is influenced and stabilized by the surrounding Myo6 sequence which guides the structural properties and features it must adopt in the context of FLMyo6. We believe that the absence of CBD^c might have led Shang et al. to report unnatural interactions involving the residues marked with green spheres in Rebuttal Table 1; Rebuttal Figures 6 and 7. In our hands only the interactions involving the residues marked by red and purple spheres could be validated in the context of the complete CBD^(N+C).

Adaptor	Myo6 Tail	K _d	n
GipC1 _{GH2 255-end}	YFP ^c CBD	164 ± 51 nM	2
GipC1 _{GH2 255-end}	YFP ^c CBD W1092A	686 ± 320 nM	1
GipC1 _{GH2 255-end}	YFP ^c CBD Y1089A	472 ± 180 nM	1
GipC1 _{GH2 255-end} E282R +D310R	YFP ^c CBD	151 ± 134 nM	2
GipC1 _{GH2 255-end} D280A	YFP ^c CBD	402 ± 101 nM	1
GipC1 _{GH2 255-end} I278A	YFP ^c CBD	556 ± 94 nM	1
GipC1 _{GH2 255-end} F319L	YFP ^c CBD	5.8 ± 2.4 μM	2
GipC1 _{GH2 255-end} G323L	YFP ^c CBD	27 ± 10 nM	1
GipC1 _{GH2 255-end}	YFP ^c CBD R1063A	30 ± 18 nM	2

Rebuttal Table 1 – Dissociation constant (K_D) ± K_D confidence (with a 68% confidence using the NTAanalysis software) determined by microscale thermophoresis of GIPC1 GH2 domain against Myo6 Tail constructs. Mutants in green do not drastically influence the interaction of the Myo6 CBD with GIPC1. In contrast, the interaction drastically drops with the red mutant. And the purple mutations increase the affinity of Myo6 CBD for GIPC1. The red and purple residues are thus part of the real main

interface of GIPC1 with the Myo6 CBD. The experiments were carried out as described in the Method section of the paper describing the microscale thermophoresis experiments with partners.

Rebuttal Figure 6 – Interactions between GIPC1 (yellow) and two CBDⁿ (dark blue and light blue) as proposed by a previous crystal structure (PDB 5V6E; Shang et al. 2017). Only the small CBDⁿ fragment was included for investigating the GIPC1 / Myo6 interaction in this earlier publication. The crystal structure proposed two interacting surfaces for GIPC1 (yellow) with two Myo6 CBDⁿ fragments (dark blue, light blue). Our MST data is compatible with only one of the two proposed interfaces (that with the CBDⁿ in dark blue). Mutations of the residues represented in green do not significantly impair the ability of Myo6 to bind GIPC1, in the context of an MST experiment including

the whole CBD (**Rebuttal Table 1**), suggesting that these residues are not part of a Myo6-GIPC1 interface. In contrast, mutation of F319 (red) to a Leucine results in a significant loss of interaction highlighting the importance of GIPC1 F319 residue for Myo6 binding. Additionally, two mutants: either R1063A (on Myo6) or G323L (on GIPC1) (purple) impact the affinity of Myo6 for GIPC1. Interestingly, these two mutations improve the affinity of GIPC1 for the Myo6 CBD.

Rebuttal Figure 7 – Two previously proposed interfaces between GIPC1 and Myo6 CBDⁿ (PDB 5V6E). **On the left:** close-up of the interface involving the GIPC1 residues labeled in green in Rebuttal Figure 6 and Rebuttal Table 1. This interface is not sensitive to mutations in its core and cannot explain the improvement in affinity when distant residues, labeled in red (GIPC1 F319) and magenta (GIPC1 G323, Myo6 R1063), are mutated. **On the right:** the second interface found in the same crystal structure, with Myo6 CBDⁿ shown in white (the same as the dark blue fragment from Rebuttal Figure 6). This interface seems to be the biologically relevant one, as it is consistent with our MST results using the complete CBD, the residues in red and magenta (which drastically affect binding) now sit at the center of the interface, and the residues in orange and green (which do not impact binding) locate to outside of the interface. C-alphas of the selected residues are shown as spheres.

The affinity results using CBD are not consistent with the formation of large oligomers previously proposed from Myo6 CBDⁿ and GIPC1 experiments. These are only possible using a severely truncated Myo6 construct. In the context of a longer Myo6 fragment, we propose that the Myo6 CBDⁿ interface (shown here in green) is prevented from interactions with GIPC1 due to CBDⁿ interactions with the rest of the Myo6 molecule, preventing its ability to interact with GIPC1.

In summary, while we agree with the reviewer that the mechanism by which GIPC1 promotes dimerization requires additional studies that are beyond the scope of this paper, they are not trivial experiments.

Our functional and structural studies clearly demonstrate:

- which Myo6 region allows dimerization
- the impact of dimerization on endocytosis
- the role of GIPC1 to promote dimerization

Further experiments, including determination of the cryo-EM structure of FLMyo6 as well as additional structural data of GIPC1 bound to a large region of the Myo6 tail will be needed to clarify how GIPC1 binding promotes dimerization.

Minor comments for the authors:

(1) The supplementary text file should include page numbers.

The page numbers have been added to the supplementary text file.

(2) Page 2, line 68, “Myo10” should be changed to “Myo7a”.

We thank the reviewer for this remark. We indeed wanted to talk about Myo10 but the reference numbers were incorrect. We thus correct this mistake by replacing reference numbers 10, 12 to 11, 13 (**Main text I. 68**).

(3) Page 4, the “CBC” in Figure 1G should be changed to “CBD”; the “CDB” in Figure 1H should be changed to “CDB”; the distance label “17.5 Å and 3.8 Å” in Figure 1H should be defined.

We thank the reviewer for pointing out these typography mistakes. We corrected **Fig. 1** accordingly. Additionally, the following sentence was added to the legend of **Fig. 1H** to define the distances labeled “17.5 Å and 3.8 Å”:

Main text I.162-163: *The distances between the C-terminus of Jo and the N-terminus of Myo6; and between the C-terminus of Myo6 and the N-terminus of In are indicated.*

(4) The panel label B and C in the supplementary Figure 4 should be revised, and the corresponding supplementary figure citation in Page 6 of the main text should be also changed.

The panel labels B and C of **Supplementary Figure 5** (named Sup Fig. 4 in the initial version of the manuscript) were exchanged (**page 8**), and the main text **page 6: Main text I.236-240** has been modified accordingly.

(5) Page 12, in the left panel of Figure 5A, the N- and C-terminal of Myo6(875-940) should be labeled and indicated. Furthermore, the dimerization interface of Myo6(875-940) in the right panel of Figure 5A

should be carefully analyzed and the interface residues should be fully labeled. The side chains of some residues that are not involved in the interactions should be removed.

The figure has been changed. We added labels for the N-terminus and C-terminus (left panel) and we labeled all the residues within the interaction interface (right panel) to clarify the left panel of Fig. 5A.

(6) In the supplementary Figure 5, the protein marker columns should be indicated, and the corresponding arrows for different protein bands should be marked correctly. In addition, for the protein labels, the “147kDa, 33kDa, 17kDa, 12kDa” should be changed to “147 kDa, 33 kDa, 17 kDa, 12 kDa”.

We thank the reviewer for this comment.

The changes were introduced to **Supplementary Fig. 6 page 10** (named Sup Fig. 5 in the initial version of the manuscript).

(7) Page 11, line 379, “polar contacts” should be changed to “polar interactions”.

We changed “polar contacts” to “polar interactions”, as requested.

Main text I.421-422: six polar interactions involving R892 with D900, and T888 with S906 (via a water molecule)

(8) Page 13, line 408, “their ability” should be changed to “their abilities”; line 410, “disturb the bundle” should be changed to “disturb the bundle formation”.

The text has been changed. It now reads as follows:

Main text I.425-428: *Three mutations (T888D.R892E.V903D) were introduced into the 875-940 peptide to assess the impact on proximal dimerization. Importantly, these residues were chosen on the surface of the 3HB so that the mutations would not disrupt the 3HB stability (Fig. 5C).*

Reviewer #4 (Remarks to the Author)

This paper investigates several structural aspects of myosin six motor protein and how it is regulated by several of its partner proteins. The paper should be revised as explained below.

We thank reviewer 4 for his/her thoughtful remarks that helped improving the clarity of the manuscript.

(1) The paper includes many acronyms that should be defined and explained.

We thank the reviewer for having identified that problem. We reviewed all the acronyms used, and modified the text for those that clearly needed a definition:

- “HMM” was removed from the manuscript. We are using “zippered dimer” instead.
- The definition of “CBD” was added in the **main text I. 85**.
- The definition of “MD” was added in the **main text I. 87**.
- The definition of “PPS” was added in **main text I. 678**, referring to the available structure 4ANJ, and the proper citation.
- The definition of “DSSO” is also provided in **main text I. 284**.

Additionally, we propose a cleaner version of Fig. 1A so that readers can more easily find the constructs mentioned in the text.

(2) The SEC-SAXS analysis is insufficient.

The authors did not show most of the data and analysis. The units in Supporting figure 1C are incorrect. The correct way is to show the original scattering data (I vs. q) at several points along the elution curve and then model the original scattering curves. In the current data presentation, it is hard to see what is going on, even if the gyration radius (R_g) is the relevant analysis and the analysis was done for the relevant q-range. The author should present and analyze their data as was done in the following examples:

<https://scripts.iucr.org/cgi-bin/paper?vg5038>

<https://scripts.iucr.org/cgi-bin/paper?jv5008>

<https://onlinelibrary.wiley.com/doi/full/10.1002/pro.4237>

We thank the reviewer for these comments. We have added all this information in Sup Data 1,3. The elution profiles and the original scattering curves at several points along the elution curve for each FLMyo6 construct and condition investigated (Sup Data 1, pages 26-27), including for the Jo-Myo6-In construct (Sup. Data 3A, page 30) (see below) are now included in Supplementary Information.

Supplementary Data 1 – FLMYo6 scattering data

SEC-SAXS data indicate FLMYo6 rearrangements depending on the presence of salt, nucleotide, or mutations. (Left) SEC elution profile of FLMYo6, the normalized absorbance (at 280 nm) is plotted as solid line, the normalized scattering intensity per frame is plotted as dotted line and the radius of gyration (R_g) per frame are plotted as black squares. Note that in SEC-MALS all the constructs tested here elute as monomers (Sup Fig. 2A) (Right) Scattering profiles ($\log(I)$ vs q) for several frames along the peak (highlighted as colored dots). **(A)** SEC-SAXS data for FLMYo6 (wild-type) in ADP.AIF₄ buffer (see Methods). The peak is symmetric and the R_g is very regular along the peak. **(B)** SEC-SAXS data for FLMYo6 (wild-type) in NF-high salt buffer (see Methods). FLMYo6 elutes earlier in NF/high salt condition than in FLMYo6 (wild-type) in ADP.AIF₄ condition (as also seen in SEC-MALS, Sup Fig. 1A). Within the first part of the peak, the R_g is pretty regular along the peak then drops by ~ 20 Å between

0.55 and 0.6 column volume (CV). This possibly indicates the presence of some FLMyo6 that would be only partially opened or possibly closed even in NF/high salt conditions. To characterize the open FLMyo6, only the frames corresponding to the first part of the peak have been considered for R_g calculation and Kratky plot generation. **(C)** SEC-SAXS data for FLMyo6 (SAHmimic) in ADP.AIF₄ buffer (see Methods). The peak elutes earlier for the mutant than for the wild-type Myo6 (as also seen in SEC-MALS, Sup Fig. 1A). The R_g is stable at the center of the peak and less precise at the edges. Thus, to characterize the open FLMyo6, only the frames corresponding to the center of the peak have been considered for R_g calculation and Kratky plot generation. **(D)** SEC-SAXS data for FLMyo6 (L926Q) in ADP.AIF₄ buffer (see Methods). The peaks elute earlier for the mutant than for the wild-type Myo6 (as also seen in SEC-MALS, Sup Fig. 1A). Along the main peak, the R_g is around ~ 130 Å as for the SAHmimic mutant. But here we note the presence of an additional peak between 0.4 and 0.45 CV with a high scattering intensity and a very high and unstable R_g . The peak very likely corresponds to some aggregation. It thus was not considered for R_g calculation and Kratky plot generation. Please note that the aggregation peak has a very low absorbance at 280 nm indicating that only a minor part of the protein is aggregated.

Supplementary Data 3 – Jo-Myo6-In scattering data

(A) SEC-SAXS data for Jo-Myo6-In in ADP.AIF₄ buffer (see Methods). **(Left)** SEC elution profile of Jo-Myo6-In, the normalized absorbance (at 280 nm) is plotted as solid line, the normalized scattering intensity per frame is plotted as dotted line and the radius of gyration (R_g) per frame are plotted as black squares. **(Right)** Scattering profiles ($\log(I)$ vs q) for several frames along the peak (highlighted as colored dots)

The Supplementary Data are cited as follows in the main text:

Main text I.116-120: Size Exclusion Chromatography coupled with Multi-Angle Light Scattering (SEC-MALS) and SEC coupled with Small-Angle X-ray Scattering (SEC-SAXS) experiments (Sup Fig. 1A-C; Fig. 1B) indicate that FLMyo6 adopts a compact conformation in the presence of ADP.P_i analogs (Radius of gyration (R_g) = 49.23 ± 0.92 (standard deviation; SD) Å) (Fig. 1B-C; Sup Fig. 1C; Sup Data 1) even at high salt concentration (~ 425 mM NaCl) (Sup Fig. 1A-B).

Main text I.164-167: To show that the fusion does not disrupt the Myo6 back-folding, we confirmed that the Jo-Myo6-In heavy chain could bind to two CaM using SDS-PAGE and that the Jo-Myo6-In behaved as a compact folded protein, even in NF/high salt condition using SEC-MALS and SEC-SAXS (Sup Fig. 2C-D; Sup Data 3A).

And as follows in the Supplementary Information:

Supplementary I.50: See scattering profiles in Sup Data 1.

Supplementary I.79: See scattering profiles in Sup Data 3.

(3) The analysis in Sup Fig 1B is unclear. Please provide more details. How were the values obtained?

We thank the reviewer 4 for this question. The first version of the plot was derived from a linear, standard normalization that was not taking in account the logarithmic relationship between Rh and retention time in SEC. Because the retention time shifts are small as compared to the total volume of the column, the new version of the plot is very similar to the first one, but it is more coherent with the expected behavior of molecules in SEC. We added a more detailed explanation in the legend, as well as the equation that was used to calculate the percentage of open FLMyo6 in each condition.

The following sentences now replace the legend of Sup Fig. 1B (**Supplementary I. 38-47**):

Initial text:

(B) Percentage of open FLMyo6 as a function of [NaCl] in ADP.VO₄ or NF conditions, as determined by SEC-MALS. The earliest and latest elution times for Myo6 monomers were taken as 100% and 0% open, respectively.

Modified text:

(B) Percentage of open FLMyo6 as a function of [NaCl] in NF and ADP.VO₄. Elution times from the two series in Sup. Fig 1A were normalized as follows: (1) the longest retention time of the entire series (180 mM NaCl in NF) was defined as 0% open FLMyo6, and the shortest retention time (800 mM NaCl in NF), as 100% open FLMyo6; (2) V_e/V_o was calculated for each data point; (3) the shifts of V_e/V_o from the minimum were employed as powers of 10 for coherence with the expected relationship between retention time and Rh in SEC; (4) Percentage of closed FLMyo6 was calculated by dividing each shift by the complete range, also as a power of 10; (5) Percentage of open FLMyo6 was calculated by subtracting %closed from one. The entire calculation can be summarized by the following equation:

$$\%open = 100 * \left(1 - \left(\frac{10^{\frac{V_e}{V_o}} - 10^{\frac{V_e}{V_o^{min}}}}{10^{\frac{V_e}{V_o^{max}}} - 10^{\frac{V_e}{V_o^{min}}}} \right) \right)$$

(4) The Y-axis legend is incorrect in Figure 1B and Sup Fig 2B. It should be as in the caption of Sup Fig 2B.

We thank the reviewer for this remark. The Y-axis legend has been corrected for Fig. 1B (Main text, page 4), Fig. 2B (Main text, page 7) and Sup Fig. 2C (named Sup Fig. 2B in the initial version of the manuscript) (Supplementary, page 4).

(5) Sup Fig 2D: Please show raw data, provide all the details about the stopped-flow experiment, present data analysis, explain it, and only then show the summarizing table.

We removed the stopped flow data summary from Sup Fig 2 and replaced it with actin-activated ATPase data that covered a large range of actin concentrations (see Sup Fig. 2E) as this data better makes the point that the Jo-In construct has very slow activity in the absence of actin although it has a similar apparent actin affinity as that previously reported by us. Furthermore, its ATPase activity at 40 μM actin that is only slightly lower than what is reported for the FLMyo6 (wild-type) in Fig. 2D.

The new Sup Fig. 2E (page 4) presents the new actin-activated ATPase experiment as follows:

Sup Fig 2

(E) Actin-activated ATPase rate of Jo-Myo6-In as a function of actin concentration. Mean values \pm standard deviation are plotted. ATPase measurements were performed as outlined in **Methods**. The V_{max} extrapolates to $\sim 0.08 \text{ sec}^{-1}$ and the K_{ATPase} (apparent actin affinity) is $\sim 7 \mu\text{M}$ actin ($\Delta G^\circ \sim -7 \text{ kcal/mol}$).

This new actin-activated ATPase is cited in the main text as follows:

Main Text I. 168-170 Actin-activated ATPase measurements revealed a very slow steady state turnover rate for Jo-Myo6-In compared to earlier measurements on wild-type Myo6³⁸, indicating that the conformational changes required to cycle on actin were greatly slowed (Sup Fig. 2E).

(6) Figure 1C, H, and Sup Fig 2E: Show the SAXS data analysis and the fit to the entire SAXS curve. Explain how it was done and the expected error of the study; Discuss alternative models that could equally fit the data and show a cluster of such models. Finally, compute the R_g of these models and compare them with the R_g analysis in Sup Fig 1.

Ab initio models (as presented in Fig. 1C) can be misleading even for good quality data since (i) different particle shapes can produce the same scattering profile, (ii) *ab initio* models can be severely biased by the presence of aggregates and (iii) *ab initio* model algorithm performance depends on the overall shape of the particle.

Still, it appeared as an interesting approach to build a first model of the FLMyo6 off-state as (1) the data collected for FLMyo6 (wild-type) in ADP.AIF₄ buffer were of good quality data (stable R_g along the peak, no aggregation seen on the scattering curve), and as (2) the Kratky plot (Fig. 1B) revealed that FLMyo6 adopts a globular shape in ADP.AIF₄ buffer and *ab initio* reconstruction algorithm are usually more reliable for globular proteins (Volkov & Svergun, 2003; doi: [10.1107/S0021889803000268](https://doi.org/10.1107/S0021889803000268)).

We evaluated the potential ambiguity of the *ab initio* reconstruction using AMBIMETER (ATSAS suite) (which evaluates how well a library of various shapes can explain a scattering curve). The ambiguity score obtained for the SAXS data for FLMyo6 in AIF₄ is 2.083 meaning that the reconstruction can be done although care is required to perform and interpret it (Petoukhov & Svergun, 2015; doi: [10.1107/S1399004715002576](https://doi.org/10.1107/S1399004715002576)).

To obtain a reliable dummy atom model of the off-state of FLMyo6 (Fig. 1C), twenty different *ab initio* models were thus generated with GASBOR program (ATSAS suite) (Svergun *et al.*, 2001; doi: [10.1016/S0006-3495\(01\)76260-1](https://doi.org/10.1016/S0006-3495(01)76260-1)) and compared. A cluster of such models are represented in Sup Data 2B. For each model, the radius of gyration (R_g) was estimated using the FoXS server (Schneidman-

Duhovny *et al.*, 2013; doi: 10.1016/j.bpj.2013.07.020; 2016; doi: 10.1093/nar/gkw389). All of them were between 48.44 Å and 49.56 Å, thus exhibited a deviation from less than 2% from the R_g determined by a Guinier analysis (49.23 Å; Sup Fig. 1). Moreover, D_{max} were around 177-185 Å for all models, in good agreement with value of the P(r) function ($D_{max}=182$ Å) (Sup Data 2A). Additionally, among the twenty models generated, DAMCLUST (Petoukhov *et al.*, 2012; doi: 10.1107/S0021889812007662) identified only one cluster. All models were thus averaged using DAMAVER (Volkov & Svergun, 2003; doi: 10.1107/S0021889803000268) to calculate the *ab initio* model pictured in Fig. 1C. Finally, manual superimposition of DAMAVER model with the EM density map of the Jo-Myo6-In shows that the size and overall shape of the DAMAVER model is rather consistent with the ones of the negative staining 3D reconstruction. Taken together, all these parameters point out to a reliable reconstruction of the Myo6 off-state.

Thanks to our SAXS *ab initio* model, a first atomic model of the off-state of Myo6 was proposed using available crystallographic structure of Myo6 pre-powerstroke motor domain, lever arm and lever arm extension (Fig. 1C). Then this model was completed with experimental atomic models of the SAH, the CBD^c and of a portion of the CBDⁿ thanks to our microscale thermophoresis, negative staining and ATPase assays (see Methods for detailed on atomic model building). As requested by Reviewer 4, we generated theoretical scattering profiles for these atomic models and compared them to the experimental scattering profiles of FLMYo6 (wild-type) in buffer containing ADP.AIF₄ (Sup Data 2D-F). The comparison was carried out using CRY SOL (Franke *et al.*, 2017; 10.1107/S1600576717007786). We obtained a very poor fitting for the pre-powerstroke MD^{Ins2/IQ/3HB} alone (Sup Data. 2D). This is not surprising since this model is largely incomplete as it lacks the full Myo6 Tail. The fitting is largely improved by the addition of the SAH and CBD^c domain to the model (Sup Data 2E) and was even better after addition of the CBDⁿ to the model (Sup Data 3F). It is important to note that even for the latter model, the structure of the Myo6 off-state remained partially incomplete as the atomic structure of the distal tail and of a portion of the CBD are currently unknown. In this context, it is thus challenging to reach a very high-quality fitting (with $\chi^2 \sim 1$). Sup Data 2 (Supplementary, page 28-29) was added to the manuscript to illustrate this analysis.

This Sup Data is cited in the main text as follows:

Main text I.149-151: (C) Representation of the *ab initio* SAXS envelope of Myo6 in ADP.AIF₄ condition (green) with MD^{Ins2-IQ-3HB} docked. Myo6 adopts a compact conformation that requires Myo6 to fold back after the 3HB domain (see Methods and Sup Data 2A-B).

Main text I.156-157: Negative staining 3D-reconstruction and the *ab initio* SAXS envelope exhibit similar overall size and shape for Myo6 (Sup Data 2C).

Main text I.171-173: The 3D reconstruction of the Myo6 off-state at ~17 Å resolution (Fig. 1F, Sup Movie 1) is consistent in shape and dimensions with SAXS data of FLMYo6 (Fig. 1C, Sup Data 2C).

Main text I.284-286: Placement of elements of the Myo6 Tail within the model improved the fitting between our atomic model and the SAXS data (Sup Data 2D-F and Sup Data 3B-C).

Main text I.693-696: The complete model of the Myo6 off-state is presented in Fig. 3A. Placement of the different domains is further supported by the crosslink experiment presented in Sup Fig. 4 and Sup Table 1. Its ability to fit our off-state Myo6 SAXS data was tested (Sup Data 2D-F and Sup Data 3B-C).

Supplementary Data 2 - FLMYo6 scattering data interpretation

(A) Pair distance distribution function ($P(r)$) of FLMYo6 (wild-type) in ADP.AIF₄ buffer ($D_{\text{max}} = 182 \text{ \AA}$) used for ab initio reconstructions. **(B)** From this $P(r)$ function, 20 ab initio models were generated using GASBOR¹⁴. A cluster of five of these models is shown in blue as examples. The final envelope is presented in Fig. 1C (obtained by averaging all the GASBOR generated models using DAMAVER¹⁵) and is also plotted in grey for comparison. All the models exhibit a similar shape; (out of the 20 models generated, DAMCLUST¹⁶ only identified one cluster of models). Moreover, their analysis^{17,18} shows that

all of them exhibit a D_{max} range of 177-185 Å and a R_g range of 48.44-49.56 Å, similar to the ones obtained from the scattering curve and $P(r)$ function of FLMyo6 (wild type) in AlF_4 (182 Å and 49.23 Å respectively). (C) SAXS envelope of FLMyo6 (grey) was manually docked into the negative staining 3D-reconstruction of the Jo-Myo6-In (deep blue). This comparison shows similar size and shape for both although the negative staining 3D-reconstruction displays more details and an additional blob corresponding to the Jo/In domain. (D-F) Next we compared the FLMyo6 off-state atomic model built based on the stained EM data from the Jo-Myo6-In construct to these SAXS measurements. Using CRY SOL¹², the theoretical scattering curves of the Myo6 off-state atomic models (red) were generated and compared to the experimental SAXS scattering curve of FLMyo6 (wild-type) in $ADP.AlF_4$ buffer (green). Residuals are plotted in the bottom part of the figure. (D) The $MD^{Ins2/IQ/3HB}$ model (as seen in Fig. 1C) fits the experimental data very poorly certainly due to the absence of the Myo6 tail in this model (E-F) Addition of the SAH, CBD^c and CBD^n domains improves the fitting to the experimental scattering data.

Similarly, a very poor fit was obtained for the nucleotide-free models presented in Sup Fig. 2F (named Sup Fig. 2E in the initial version of the manuscript) (see Rebuttal Figure 8 below). This poor fit is not surprising as the Myo6 lever arm extends when no nucleotide is bound to the motor domain and the Tail are missing in this model. Please note that contrary to the pre-powerstroke $MD^{Ins2/IQ/3HB}$ model, adding the lever arm extension and the tail in the continuity of the lever arm would not allow to obtain a model in which the motor and the tail would interact. Sup Fig. 2F shows that a nucleotide-free conformation of the motor would position the lever arm orientation such as creating a clash with the Tail.

Rebuttal Figure 8 – Fitting of the nucleotide-free Myo6 models presented in Sup Fig. 2F.

Using CRY SOL (Franke et al., 2017; 10.1107/S1600576717007786), the theoretical scattering curves of the two Myo6 $MD^{Ins2/IQ}$ /Jo-In models presented in Sup Fig. 2F were generated and compared to the experimental scattering curve of Jo-Myo6-In in $ADP.AlF_4$ buffer (green). The theoretical scattering curve of the model represented in the left part of Sup Fig. 2F (with the lever arm partially out of the density) is plotted in red and the theoretical scattering curve of the model represented in the right part of Sup Fig. 2F (with the lever arm directed toward the Jo/In domain) is plotted in black. Both models fit the data very poorly with $\chi^2 > 350$. Residuals are plotted in the bottom part of the figure.

(7) Based on the density map, compute the expected SAXS curve and compare it with the SAXS data and the earlier analysis.

We have not been able to find the program required to compute the scattering curve from a negative staining density map. Instead, we computed expected scattering curve from the Jo-Myo6-In model

built based on the negative staining density map (shown in Fig. 3A) and compared it to the SAXS data collected for the Jo-Myo6-In model (see our answer to the following question of this Reviewer).

(8) Then compute and add the expected SAXS curve from each PDB, compare it with the SAXS data, and then discuss the differences.

The expected SAXS curves were computed for each Myo6 off-state models proposed along the paper. MD^{Ins2/IQ/3HB}, FLMyo6 without CBDⁿ and FLMyo6 models are presented in Sup Data 2 and discussed in our answer to question 6 of Reviewer #4. To complete this analysis, the expected SAXS curve of the final Jo-Myo6-In model (as presented in Fig. 3A) was generated using CRY SOL (Franke et al., 2017; 10.1107/S1600576717007786). Comparison with the experimental scattering curve of Jo-Myo6-In in ADP.AIF₄ buffer gave a poor fit with $\chi^2=113.9$ (Sup Data 3B). To improve the fit, we used the SREFLEX program (Panjkovich & Svergun, 2016; doi: 10.1039/c5cp04540a) to refine the model against the experimental SAXS data. After refinement, the fitting is much better with $\chi^2=9.78$ (Sup Data 3B). By superimposing the Jo-Myo6-In model before and after refinement by SREFLEX, we observed that only the Jo/In domain was displaced (Sup Data 3C), while the rest of the model remained unchanged. This difference suggests the expected ability of the Jo/In subdomain to vibrate with respect to the FLMyo6 structure. The Jo/In entity adopts different positions relative to the FLMyo6 entity. As for the FLMyo6 model we could build, it is important to note that the Jo-Myo6-In model remains partially incomplete as the atomic structure of the distal tail and that of a portion of the CBD are currently unknown. In this context, it is thus challenging to reach a very high-quality fitting (with $\chi^2\sim 1$).

Sup Data 3B-C are now cited in the main text as follows:

Main text I.284-286: *Placement of elements of the Myo6 Tail within the model improved the fitting between our atomic model and the SAXS data (Sup Data 2D-F and Sup Data 3B-C).*

Main text I.693-696: *The complete model of the Myo6 off-state is presented in Fig. 3A. Placement of the different domains is further supported by the crosslink experiment presented in Sup Fig. 4 and Sup Table 1. Its ability to fit our off-state Myo6 SAXS data was tested (Sup Data 2D-F and Sup Data 3B-C).*

Supplementary Data 3 – Jo-Myo6-In scattering data

(B) Using CRY SOL¹², the theoretical scattering curves of the Jo-Myo6-In model were generated before (red, left) and after (black, right) model refinement using SREFLEX¹³. These curves were compared to the experimental scattering curve of Jo-Myo6-In (yellow). Residuals are plotted in the bottom part of the figure. The Jo-Myo6-In model fit is improved after SREFLEX refinement. **(C)** Superimposition of the Jo-Myo6-In model before and after refinement by SREFLEX. The Jo/In domain is the only part of the model to be displaced, the rest of the model remains mostly unchanged. This highlights the ability of the Jo/In domain to vibrate with respect to Myo6.

(9) Why was cryo-TEM not done? A negative stain is less accurate. Please do cryo-TEM single particle analysis.

We completely agree with reviewer #4 that Single Particle Analysis by Cryo-TEM can produce maps with much better resolution. This would be an ideal way to see details of the model that are essential to describe the different interfaces involved in stabilizing the Myo6 off-state. However, SPA of small and flexible proteins such as FLMyo6 is a challenge in itself.

We have started experiments in this direction but cryo-EM studies of FLMyo6, like for many molecular motors are not trivial due to the flexibility of the motor in the off-state. In addition, in the case of

FLMyo6, interpretation of the maps would require determining high resolution as part of the structures for part of the sequence is unknown, and in fact is likely to be influenced by the environment found in the off-state of the motor. The lack of structural data for metamorphic structural parts of Myo6 whose flexibility may also allow different conformations to be explored indicates that the determination of this structure is a challenge, even with current cryo-EM approaches.

We do not consider this structure as being essential for this story. To ascertain of the validity of our low resolution FLMyo6 model, we carried out a crosslinking mass spectrometry analysis of the FLMyo6 using the disuccinimidyl sulfoxide (DSSO) crosslinker (see **Supplementary Text** (page 21), **Sup Table 1** (page 22-23) and **Sup Fig. 4A-C** (page 7)). The results are consistent with our model, reinforcing the main conclusions we have proposed from our negative stain model.

(10) The K_D analysis is not explained in Sup Fig 3. Many more details are required. The legends are not explained, and the Y-axis units are not defined. The error analysis and the model to which the data fit should be described.

According to the reviewer's request, the legend of **Sup Fig 3** was completed as follows:

Main text I.96-104: *Microscale thermophoresis profiles and fits corresponding to the data exposed in Table 1. The baseline corrected normalized fluorescence (ΔF_{norm}) is plotted against the concentration of Myo6 Head constructs. Myo6 Tail construct was added in a constant concentration (100 nM) in all our assays (see Methods). K_D was determined with a confidence of 68 % using the NTAAnalysis software. The error bars are for the standard deviation. A standard fitting model derived from the law of mass action was used:*

$$f(\text{concentration}) = \text{Unbound} + \frac{(\text{Bound} - \text{Unbound})([\text{Myo6 Head}] + [\text{Myo6 Tail}] + K_D - \sqrt{([\text{Myo6 Head}] + [\text{Myo6 Tail}] + K_D)^2 - 4[\text{Myo6 Head}][\text{Myo6 Tail}]})}{2[\text{Myo6 Tail}]}$$

(with Unbound corresponding to the response value of the unbound state and bound corresponding to the response value of the bound state)⁵.

(11) The authors should calculate the standard Gibbs free energies associated with their K_D values and add them to Table 1. Please do it for other K_D values.

We thank Reviewer #4 for this suggestion. Values were added in kcal/mol in **Table 1 (Main text I. 202)**, as well as on **Main text I. 411**, and **Supplementary I. 86-87**.

REVIEWERS' COMMENTS

Reviewer #1 (Remarks to the Author):

I am satisfied with the authors response to my criticisms

Reviewer #3 (Remarks to the Author):

The authors have addressed all my points and concerns. Now, I have no further comments. With the additional data, the manuscript is much stronger.

Reviewer #4 (Remarks to the Author):

The authors significantly improved the manuscript and it can be published.